# The PECAn image and statistical analysis pipeline identifies Minute cell competition genes and features

Michael E. Baumgartner[1,3,6] ✉, Paul F. Langton [1,6], Remi Logeay[1], Alex Mastrogiannopoulos [1], Anna Nilsson-Takeuchi[1,4], Iwo Kucinski [2,5], Jules Lavalou [1] & Eugenia Piddini [1] ✉

Investigating organ biology often requires methodologies to induce genetically distinct clones within a living tissue. However, the 3D nature of clones makes sample image analysis challenging and slow, limiting the amount of information that can be extracted manually. Here we develop PECAn, a pipeline for image processing and statistical data analysis of complex multi-genotype 3D images. PECAn includes data handling, machine-learning-enabled segmentation, multivariant statistical analysis, and graph generation. This enables researchers to perform rigorous analyses rapidly and at scale, without requiring programming skills. We demonstrate the power of this pipeline by applying it to the study of Minute cell competition. We find an unappreciated sexual dimorphism in Minute cell growth in competing wing discs and identify, by statistical regression analysis, tissue parameters that model and correlate with competitive death. Furthermore, using PECAn, we identify several genes with a role in cell competition by conducting an RNAi-based screen.

With the advent of safe, non-invasive tools for generating and marking genetically distinct subpopulations of cells within intact organisms, clonal analysis has become a common tool for the study of heterogeneous cell populations and mosaic tissues (Reviewed in[1]). These tools come in many forms but achieve a common goal: a subset of cells within a tissue acquire a genetic alteration absent in the surrounding tissue, creating genetically distinct territories of cells.

While this technique presents unique avenues for investigating tissue biology, extracting information from mosaic tissues by microscopic analyses is often a bottleneck. The resulting images, typically acquired by 3D confocal microscopy, are information rich and provide insights on phenotypes, such as signal intensity of reporters, clone numbers, clone size, shape and apoptosis. Extracting this information, however, is time consuming and complicated. In recent years, a number of powerful image analysis tools have been developed, including image analysis environments enabling construction of custom algorithms, such as CellProfiler[2], and more targeted software, such as tools for analysis of morphogenesis and cell shape in situ[3–5] and machine-learning enabled segmentation of twin-spot clones[6]. However, there are no image analysis tools designed to analyse complex, multi-genotype 3D image stacks and to measure, in the whole image or in specifically marked subregions, diverse parameters such as levels of apoptosis, fluorescence and speckle intensity, cell number and density. Analysis of such confocal images is therefore typically performed in a manual or semi-automated fashion. Performing such analysis by hand is time-consuming, inconsistent, and error prone, with large datasets often requiring days of work. This bottleneck therefore impairs both the speed, quality, and scope of data generation.

[1]School of Cellular and Molecular Medicine, University of Bristol, Biomedical Sciences Building, University Walk, Bristol BS8 1TD, UK. [2]The Wellcome Trust/ Cancer Research UK Gurdon Institute and Zoology Department, University of Cambridge, Tennis Court Road, Cambridge CB2 1QN, UK. [3]Present address: Perelman School of Medicine, University of Pennsylvania, 3400 Civic Center Blvd, Philadelphia, PA 19104, USA. [4]Present address: Cancer Sciences, Faculty of Medicine, University of Southampton, Southampton, UK. [5]Present address: Wellcome & MRC Cambridge Stem Cell Institute and Department of Haematology, University of Cambridge, Cambridge, UK. [6]These authors contributed equally: Michael E. Baumgartner, Paul F. Langton. ✉e-mail: michael.baumgartner@pennmedicine.upenn.edu; eugenia.piddini@bristol.ac.uk

An example of this challenge is demonstrated by imaging experiments used for the study of Minute cell competition in *Drosophila* imaginal wing discs – a process whereby cells carrying heterozygous mutations in ribosomal protein genes (Rp/+), known as *Minutes*, are eliminated from mosaic tissues when proximal to wildtype cells[7] (Supplementary Fig. 1). *Minute* cells are therefore said to act as 'losers' relative to wildtype 'winners.' Such experiments require careful localization of dying cells and categorization by genotype. In the field of cell competition, in particular, there is a pressing need for automated image and single cell analytic techniques[8].

In this work, we present a comprehensive, high-throughput image and data processing pipeline for automating analysis of mosaic experiments: the Pipeline for Enhanced Clonal Analysis (PECAn). This software readily performs myriad clonal analyses, including identifying the number, size, position, and fluorescence properties of single cells and cell territories by region and genotype in 3D space. In addition, PECAn features an incorporated statistical analysis and graph generation application, allowing users to visualize and evaluate their results in an automated fashion and at scale. This software is built to be flexible, user friendly, and accessible to biologists with no computational or image analysis background, as all inputs are made using graphical user interfaces and require no prior programming knowledge. In this study the use of PECAn applied to the study of *Minute* cell competition allowed us to: identify an unappreciated sexual dimorphism in growth patterns in competing discs, characterize rigorously *Rp/+* cell death properties in competing and non-competing tissues, identify by logistic regression analysis parameters that accurately model and correlate with competitive cell death, and carry out a highly sensitive targeted RNAi-based genetic screen for modulators of Minute cell competition. These applications demonstrate that PECAn reduces operator induced variability and improves speed, consistency, and sensitivity relative to existing techniques, while allowing for many useful experimental analyses.

## Results

### PECAn design strategy

The PECAn image analysis pipeline consists of two complementary components: (1) a FIJI/imageJ[9] plugin for analysing images and extracting measurements and (2) an R-Shiny-based web application for processing data, running statistical tests, and generating graphs and plots (Fig. 1a). In order to make this software as accessible to biologists as possible, this design prioritizes ease-of-use, requires no prior computational or programming knowledge, and runs entirely on free, publicly available software platforms that are familiar to biologists. All user inputs and processed outputs are made via graphical user interfaces (GUIs) with step-by-step instructions. This programme is also optimized for high-throughput batch processing and is capable of analysing large datasets of hundreds of samples. In order to maximize customizability, the pipeline incorporates WEKA machine-learning based image analysis[10]. WEKA provides, in FIJI, a GUI-enabled means of training an algorithm using supervised machine-learning, which can classify pixels according to user-defined categories. PECAn links this function directly to image segmentation and measurements. Therefore, images that can be classified by WEKA can be readily incorporated into the pipeline without writing any additional code. The pipeline furthermore incorporates tools to allow users to incorporate their own code as modules, therefore custom scripts or additional machine learning segmentation tools can be readily incorporated into the pipeline by anyone with experience in coding using FIJI-compatible languages such as Python and Java. An example analysis output, exhibiting various functionalities in a three-dimensional sample, is included in Supplemental Video 1.

### Marker-based segmentation

The first step in mosaic analysis is to segment cell populations by visible markers. Identification and separation of regions on the basis of fluorescent signal is carried out through the ImageJ/FIJI plugin, which includes built-in algorithms suitable for two-genotype classification in mosaic tissues. However, no single algorithm can properly segment all different means by which mosaic patches can be marked. This pipeline therefore integrates WEKA machine-learning enabled segmentation. Thus, PECAn can process any image of mosaic tissue, which can be segmented either by the built-in algorithm, via WEKA, or by custom user-made code. The pipeline can also process images with a large number of distinctly marked genotypes, making it suitable for assessing multi-genotype mosaic tissues generated with complex multi-colour genetic tools. To enable automated analysis of cell interactions, each marked domain is then further subdivided into a border region and a centre region (Fig 1a).

### Segmentation of fluorescently labelled tissue subdomains

The PECAn FIJI plugin allows for analysis of individual fluorescently labelled patches of tissue (Fig. 1b). To do so, the algorithm identifies each disjoint region identified via the marker-based segmentation as distinct ROIs in all Z-planes. The algorithm then compares the ROIs in three dimensions for contiguity while also accounting for instances where patches split apart or merge together across Z-planes. This allows the algorithm to accurately count the size and number of each individual patch, while also assessing them for additional fluorescence-based parameters, such as levels of foci coverage/density and regional fluorescence intensity.

### Single cell segmentation

Segmentation of individual cells operates in a similar fashion. Disjoint regions corresponding to individual cells are segmented either via built-in algorithms or via a WEKA classifier. Each ROI is then compared to ROIs on adjacent Z-planes. Individual ROIs are linked across Z-planes using centroid-based tracking. Individual cells can then be analysed by various metrics, including their position within the tissue, fluorescence intensity, distance to patch border, viability and population-level parameters of the entire tissue (Fig. 1c).

### Foci segmentation

To enable the segmentation of reporters and stainings that present as distinct foci, such as antibody stainings for cleaved caspases and TUNEL assays and other assays for cell death, we developed and embedded in PECAn an algorithm for foci segmentation, which uses alternating pixel intensity and size-based filters (Supplementary Fig. 2). Alternatively, this can be substituted with a WEKA classifier or custom code for foci detection. It is then possible to calculate foci enrichment within given regions of the image, such as Regions of Interest (ROI) identified via marker-based patch segmentation (Supplemental Video 1 and Supplementary Fig. 2). From this, the algorithm determines the area and percentage of overlap – e.g. the fraction of an ROI that is positive for the cell death reporter (Supplementary Fig. 2).

Counting the number and/or density of discrete foci in 3D, e.g. to get an estimate of the number of dying cells, presents additional challenges, as foci must be accounted for in three dimensions to prevent multiple counting of the same foci, and the algorithm must be able to distinguish between tightly packed foci. PECAn accomplishes this through a watershed-based approach using the MorphoLibJ toolkit[11], combined with 3D centroid-based tracking, allowing for precise counts and localization of foci. Individual foci can then be analysed by various parameters, such as their position within the tissue and proximity to various landmarks (Fig. 1c). This metric can furthermore be cross-referenced against individual cell counting algorithm to allow the software to determine an accurate count of cells positive for a given foci staining in each region of the 3D tissue.

### Fluorescence and speckle measurements

In addition, we incorporated fluorescence intensity and speckle analysis functionalities into this pipeline. When combined with the other

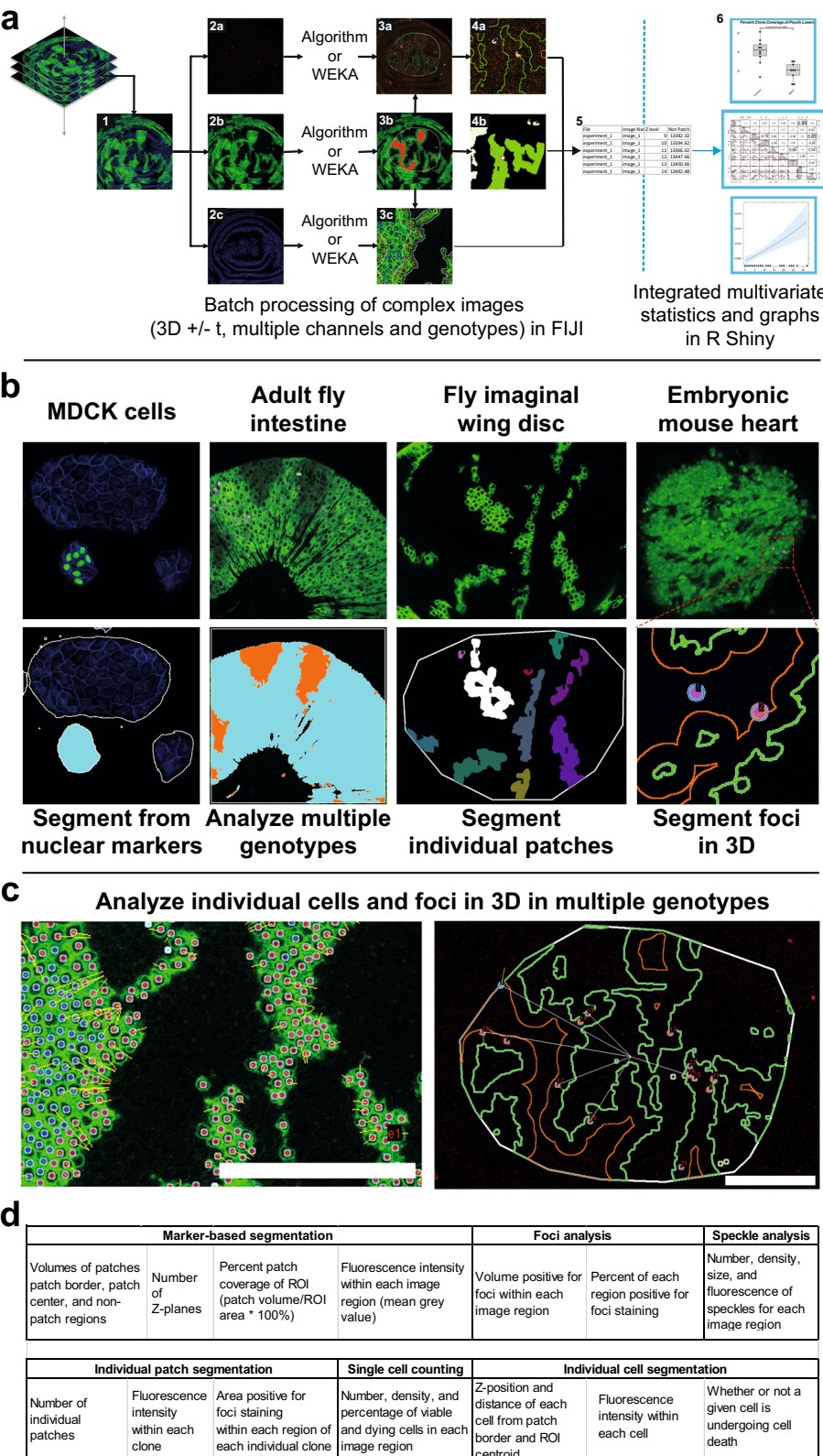

Nature Communications | (2023)14:2686                                                                                                                3

segmentation modalities present in the software, this allows for measurements of fluorescence intensity and speckle density, number, size and other parameters within different ROIs.

Altogether, this image analysis pipeline is compatible with a broad range of immunofluorescence and reporter signals, which can be analysed automatically to extract regional properties of the reporters relative to the mosaic composition of the sample. By combining these

various modalities, PECAn can rigorously and automatically quantify numerous biologically relevant phenotypes (Fig. 1d).

### R-Shiny-based web application
To facilitate and improve the next step in data analysis, i.e. statistical testing, and plot generation, we developed a companion application in R statistical software to handle the data generated from the FIJI/ImageJ

**Fig. 1 | PECAn is a versatile tool for the analysis of complex 3D images.**
**a** Structure of PECAn image analysis software. Images are initially analyzed in FIJI/ImageJ. In order to preserve three-dimensional information, each Z-plane (1) of each image is analysed sequentially and compared to adjacent Z-planes. The channels of the image—for instance foci/cell death staining (2a), cell patches (2b), and nuclear mask (2c) are split apart and processed independently either by the built-in algorithms or by a WEKA classifier. For instance, the foci/cell death mask (3a) can be used to perform three dimensional segmentation of individual foci and can then be cross referenced against the cell patch ROIs (3b) to yield a density of foci in different regions within the image (4a) and the level of cell death in individual cell patches (4b). Furthermore, cell patch and nuclear stain channels can be used together to identify individual cells in three dimensions (3c). The data are exported as CSV files (5) which can then be uploaded into the R Shiny companion app for statistical analysis and graph generation (6). **b** PECAn is able to analyse

samples derived from diverse tissues and model organisms. PECAn identifies distinct subpopulations of MDCK cells by combining a nuclear GFP marker with seeded-region growing techniques (left). PECAn analyses two distinct genotypes in three dimensions in an adult *Drosophila* intestine sample (center left). PECAn can segment and analyze individual cell patches in three dimensions in a *Drosophila* imaginal disc (center right). PECAn identifies density of TUNEL-positive foci in embryonic mouse hearts carrying differently labelled cell patches (right). **c** PECAn can identify individual cells (left) and individual foci (right) in three dimensions and analyse their position within the tissue, proximity to cell patch borders (yellow lines), or distance to the patch centroid (white lines). Individual cells/foci are colour coded; a red border denotes a cell in the border region of a patch, and a blue border denotes a cell in the centre region of a patch. Scale bars correspond to 50 μM. **d** Examples of some of the key parameters for which PECAn can assess.

plugin. While R is a powerful statistical software with excellent graphical packages, it is not user-friendly to someone without prior programming experience. We therefore incorporated a R analysis script into a Shiny web app, thus combining the statistical power of R with a user-friendly web-based GUI (Fig. 2). This application automatically processes data generated by the FIJI/ImageJ plugin but can also analyse generic datasets, saved as CSV files.

Once the data generated from the FIJI/imageJ plugin is uploaded and the user specifies the genotypes, experimental groupings, and the desired analyses, the app automatically runs appropriate statistical tests and generates output plots using a GUI accessible on any conventional web browser. The user can assess the dataset in two general ways: (1) classical uni- and bi-variate analysis or (2) multiple regression (Fig. 2). For uni- and bi-variate analysis, the user specifies which variety of graph to generate, which variable(s) to analyse, which statistical test to run (such as a *t*-test, ANOVA, or correlation coefficient), which effect size metric to run, and if/how to adjust p-values for multiple comparisons. The resulting graphs are customizable and can be exported as publication quality images using the ggplot2 package in R. For multiple regression analysis, the user selects between various techniques, such as logistic, linear, and Poisson regression, and specifies which variables to act as dependent and predictor variables. Upon running the analysis, the app automatically runs appropriate assumptions tests, produces effects plots, and generates a suite of diagnostic plots and metrics to enable the user to evaluate the quality of their analysis (Fig. 2). The app furthermore provides tools for performing data transformations. Should the user desire to run any statistical tests not supported by the app, the processed data can be exported and incorporated into any conventional statistical analysis software.

To ensure that the multiple regression functionalities were functioning properly, we ran publicly available tutorial datasets for multiple logistic regression, Poisson regression, negative binomial regression, and linear regression through the analysis app. These analyses generated the expected results (Supplementary Data 1), indicating that the statistical packages had been properly incorporated into the application.

### PECAn Fiji validation

Next, we tested PECAn discovery power, by challenging it with 3D multi-channel confocal images of *Drosophila* imaginal wing discs undergoing Minute cell competition. In Minute cell competition cells heterozygous mutant for one of several ribosome protein (*Rp*) genes behaves as 'losers' and are eliminated by wild-type cells (Supplementary Fig. 1, ref. 7). Competitive elimination results from a combination of cell intrinsic differences in tissue growth (Minute cells grow more slowly and die more frequently than wild-type cells[12–14]) and non-cell autonomous effects (*Minute* cells undergo competitively induced apoptosis when they border wild-type cells[14–16]). Minute cell competition is typically studied by creating mosaic wing discs containing wild-type and *Rp*+/− patches of tissue, making it an ideal system to test

PECAn's functionalities. Indeed, it is with this biological problem in mind that PECAn was initially created. To induce Minute cell competition, we utilized a *Drosophila* stock carrying an insertion that expresses an excisable copy of the ribosome gene *RpS3* (under control of an actin promoter) in an *RpS3*+/− mutant background (*RpS3[Plac92], act > RpS3>Gal4*)[14]. Excision of the transgene generates *RpS3*+/− cells in a genetic background wherein *RpS3* expression is rescued by the *act > RpS3* construct at levels sufficient to induce cell competition; hence we refer to this stock as Minute in Wildtype-like Organism (MiWO) (Supplementary Fig. 3a and 3c and ref. 14). Indeed the MiWO construct partially rescues the *Minute* phenotype: when comparing time to pupariation in larvae that were wildtype, *RpS3*+/−, MiWO, or MiWO harbouring *RpS3*+/− patches (Supplementary Figure 3c), the MiWO construct yielded a substantial but not complete rescue of the developmental delay seen in *RpS3*+/− larvae. Interestingly, patch induction had no detectable impact on time to pupariation in MiWO larvae.

To measure the levels of competitively induced cell death, border death vs death in the centre of the patch is used as a metric of *Minute* competition, and differences in border death between samples are used to assess the relative strength of competitive interactions on cell death[14–17]. As cell competition also results from different growth rates between winners and losers, another primary metric for assessing competition is the size of winner/loser cell patches[15]. These measurements have therefore been automated and incorporated into the PECAn pipeline.

Before using PECAn to study cell competition, we sought to validate it by challenging it in three separate ways: (1) using semi-synthetic images, (2) comparing with human analysis, and (3) testing its sensitivity at identifying known strong and mild modulators of Minute cell competition. Semi-synthetic images provide a means of testing a large programme for errors in the code and data output. In short, images with known properties are fed into the pipeline, and the outputs are compared against the known input values (Fig. 3a). The pipeline yielded results consistent with the input values across all parameters tested (Fig. 3b), indicating that the software successfully computes the right operations when producing the desired measurements. To compare the pipeline's performance against human analysis, an image dataset was both analysed by hand and run independently through PECAn. The results generated by the macro were consistent with those generated by users: both the density of apoptosis in the patch border region (Fig. 3c, d, Supplementary Fig. 3b) and the counts of individual cells (Fig. 3e, f) were not significantly different, showing that the pipeline performs comparably to manual quantification. The pipeline was then tested for its ability to identify known modulators of Minute cell competition. We expressed RNAi lines against established suppressors of cell competition, specifically *Dronc* and *Xrp1*, in loser cells using the MiWO system. *Dronc-RNAi* has previously been shown to yield a mild rescue, as it only rescues *Rp/+* cells from cell competition-induced apoptosis[15], whereas mutation in *Xrp1* yields a strong rescue, as it inhibits both competitive death and the slow growth phenotype of

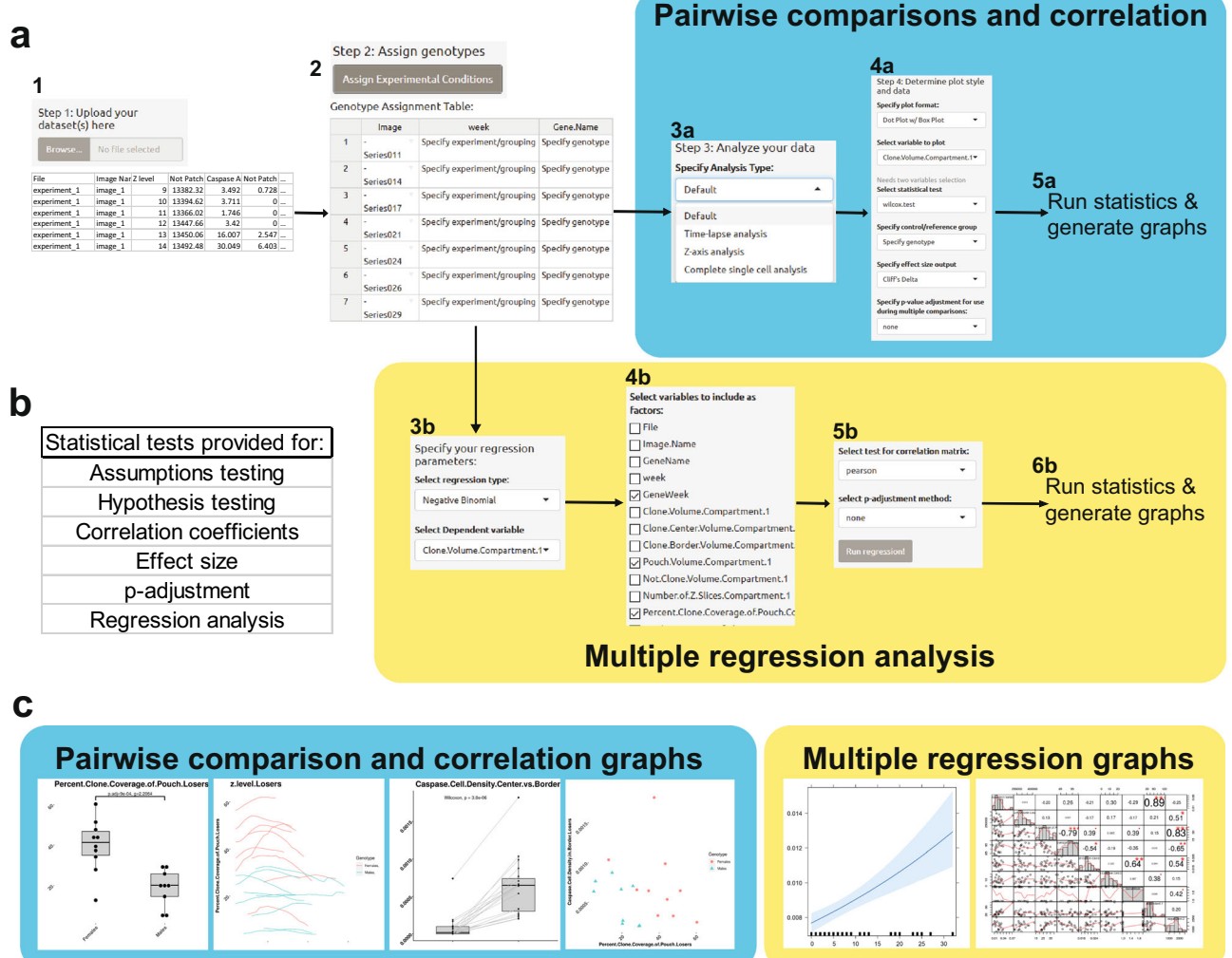

**Fig. 2 | Workflow of the PECAn R Shiny statistical analysis application. a** Users upload the datasets generated by the FIJI/ImageJ plugin (1) and group images by experiment and treatment/genotype/condition using the graphical user interface (2). The user can then assess their samples either via classical uni- and bi-variate tests (a/blue) or via multiple regression analysis (b/yellow). To perform uni- and bi-variate tests, the user specifies an analysis to perform (3a) and then selects an appropriate plot to generate, which tests to run, which reference group to use, and which (if any) p-correction for multiple comparisons to perform (4a). Upon running the analysis (5a), output graphs and statistical tests are performed, along with tests for parametric assumptions. To perform multiple regression analysis, the user specifies which form of regression to run and which parameter should act as the dependent variable (3b). The user then selects which parameters to include as predictor variables (4b), which test to use for generating a correlation matrix, and which (if any) p-correction for multiple comparisons to perform (5b). Upon running the analysis (6b), the regression is run along with test for assumptions and both output and diagnostic plots are generated. **b** A summary of the kinds of tests supported by the app. **c** Examples of plots generated by the app, using the ggplot2 package in R. Different plots are generated for uni- and bi-variate tests (left/blue) and for multiple regression analysis (right/yellow).

Minute cells[18–20]. Samples of MiWO wing discs expressing *Dronc-RNAi* or *Xrp1-RNAi* were fed into the pipeline and assessed for density of apoptosis at the $Rp^{+/-}$/wild-type border and for size of the area covered by $Rp^{+/-}$ cells. The pipeline successfully detected both known modulators and distinguished these from controls. Importantly, PECAn distinguished between the rescue modalities of these two genetic manipulations, scoring a rescue in both $Rp^{+/-}$ coverage of the pouch and competitive death for *Xrp1-RNAi* and a rescue in death but not in pouch coverage for *Dronc-RNAi*. Thus, PECAn not only allows for fast and sensitive detection of phenotypes but can also automatically provide insight into the biological parameters responsible for the rescue of Minute cell competition (Fig. 3g–i).

**PECAn reveals sexual dimorphism in growth of minute cells**
Having successfully validated PECAn, we then sought to utilize this toolset to investigate at impressive scale and sensitivity some of the parameters of Minute cell competition. Sexually dimorphic phenotypes are common in *Drosophila* cell biology[21,22], and recent work has identified a sexual dimorphism in a *Drosophila Myc* super-competition model, wherein female wing discs exhibit an apparent increased susceptibility to loser cell elimination[23], but this has not been investigated for Minute cell competition. This would be an important confounding factor when assessing Minute cell competition phenotypes, if not properly controlled. To ask whether Minute cell competition displays sexual dimorphism, we dissected separately male and female larvae from (1) a cross of wildtype males bred with MiWO females, and (2) a cross of wildtype females bred with MiWO males (Fig. 4a–c). All images were subsequently analysed with PECAn. No differences were observed between female larvae, regardless of parental genotype. Surprisingly, however, we observed a strong sexual dimorphism: $RpS3^{+/-}$ cells covered a smaller proportion of the tissue in males relative to females (Fig. 4a, b). Interestingly, this effect was confined to pouch coverage, as we did not observe differences in the frequency of cell competition-induced apoptosis between males and

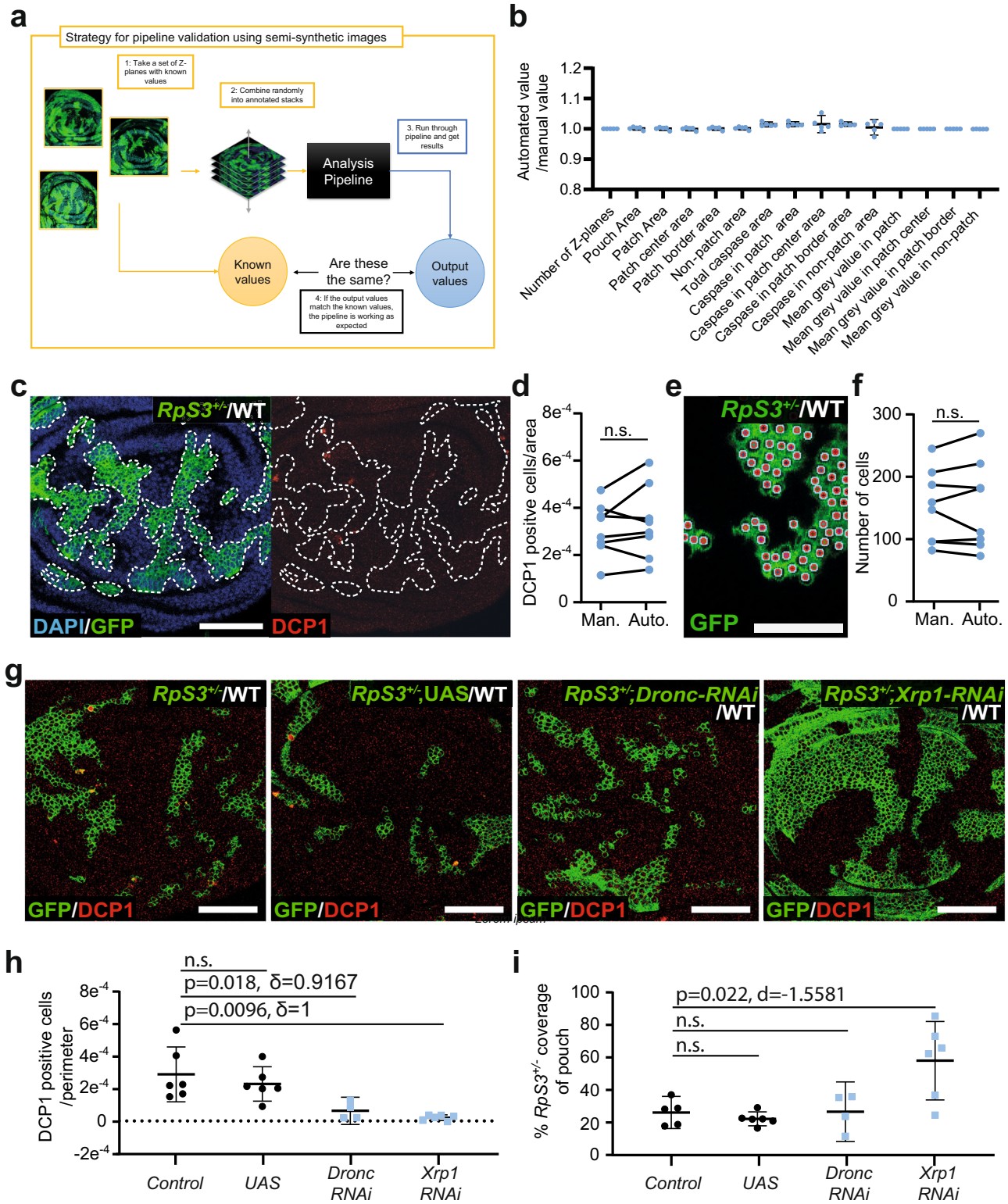

females (Fig. 4a, c), indicating that this reflects a difference in the growth rates of Minute cells in mosaic tissues. As the *hs-FLP* construct used in this experiment to induce *RpS3*^(+/−) cells is carried on the X chromosome, we repeated this assay using a *hs-FLP* construct carried on an autosome to rule out the possibility of a sex chromosome-driven difference in flippase expression. With the autosomal *hs-FLP*, we again observed a striking difference between males and females (Supplementary Figure 4a, b). To rule out the possibility that this dimorphism is due to a bias in *RpS3*^(+/−) cell induction inherent in the MiWO construct

itself, we assessed the same *RpS3* flp-out construct in a non-*Minute* background (*RpS3*^(+/+), *act > RpS3>Gal4*). We observed no difference in *RpS3*^(+/−) pouch coverage between males and females (Fig. 4d, e). To confirm that this phenotype is not specific to the MiWO system, we generated wing discs containing wildtype winners and *RpS3*^(+/−) losers using the traditional *FRT* mitotic recombination technique, and again we observed that *RpS3*^(+/−) pouch coverage was smaller in males than females (Fig. 4f, g). This dimorphism was still observed when MiWO patches were generated in larvae that were heterozygous mutant for

**Fig. 3 | Validation of the PECAn pipeline. a** Schematic of semi-synthetic image validation strategy. Z-planes with known parameters were randomly combined into Z-stacks. Known images were analysed using the macro and results were compared against known values. **b** Macro results divided by known values ($n = 5$ independent samples), with mean and 95% CI shown. No statistical test performed. All values approximate expected values, with deviations attributable to rounding errors. **c** Representative image of a competing wing disc stained with anti-cleaved Dcp-1. $RpS3^{+/-}$ patches are marked by GFP (green), and nuclei by DAPI staining (blue) (Left). The anti-cleaved Dcp-1 staining is shown (red) and the dotted line represents the outline of the loser patches (Right). **d** Comparison of the density of apoptotic events in the patch border as determined by hand against macro outputs. **e** Sample output of individual cell counts performed by PECAn in competing wing disc. Individual cells are marked with a uniquely colour-coded stamp. **f** Comparison of quantifications generated manually against those generated by the macro. **g** Representative images of wing discs harbouring competing $RpS3^{+/-}$ cells (green)

immuno-stained for cleaved-Dcp-1 (red). From left to right, the genotypes are: no transgene other than GFP, expression of an empty UAS promoter, expression of an RNAi against *Dronc*, and expression of an RNAi against *Xrp1*. **h** Automated quantification of density of cleaved-Dcp-1-positive cells in the patch border region. Measure of center and error bars are shown as mean and 95% CI. Statistics reflect two-sided Wilcoxon–Mann–Whitney $U$-test without adjustment for multiple comparisons with Cliff's δ effect size. Number of independent biological samples are: replicate 1: $n_{control} = 6$, $n_{UAS} = 6$, $n_{Dronc} = 4$, $n_{Xrp1} = 6$; replicate 2: $n_{control} = 12$, $n_{Dronc} = 15$. **i** Automated quantification of the percentage of the wing disc pouch occupied by $RpS3^{+/-}$ cells with mean and 95% CI shown. Statistics reflect two-sided $t$-test without adjustment for multiple comparisons with un-pooled Cohen's d effect size. Number of independent biological samples samples per replicate are: replicate 1: $n_{control} = 5$, $n_{UAS} = 6$, $n_{Dronc} = 4$, $n_{Xrp1} = 6$; replicate 2: $n_{control} = 12$, $n_{Dronc} = 15$. Scale bars correspond to 50 μm. Source data are provided as a Source Data file.

*Xrp1* (Supplementary Fig. 4c, d) or when loser cells expressed *Xrp1-RNAi* (Supplementary Fig. 4e, f), conditions that are known to rescue cell competition[18,19,24]. There is, however, a reduction in the effect size for the observed dimorphism in $Xrp1^{+/-}$ conditions (δ = −0.908 for $RpS3^{+/-}$, $Xrp1^{+/+}$ vs δ = −0.533 for $RpS3^{+/-}$, $Xrp1^{+/-}$). Thus, while this data suggest that $RpS3^{+/-}$ clonal sexual dimorphism does not require Xrp1, it is possible that Xrp1 may contribute to, but not fully determine, this dimorphism. To assess whether this dimorphism is generalizable to *Minute* mutations other than *RpS3*, we repeated this assay using a mutation in a different Rp gene, *RpL27A*, known to yield a Minute phenotype[25]. In order to confirm that the *RpL27A* allele used exhibited loser-associated stress pathway activation, we confirmed increased eIF2α phosphorylation (Supplementary Fig. 5a) and increased Xrp1 expression, using the *Xrp1-LacZ* reporter (Supplementary Fig. 5b) in $RpL27A^{+/-}$ cells. We furthermore assessed for proteotoxic stress by staining for ref(2)P (p62), an autophagy adaptor protein, which we have previously shown to be enriched in cells carrying $RpS3^{+/-}$ and $Mahj^{-/-}$ loser inducing mutations[14]. Consistent with this, we observed an increase in p62 signal intensity in $RpL27A^{+/-}$ cells (Supplementary Fig. 5c). This is different from a recent report that *RpL27A* heterozygote losers failed to exhibit an accumulation of p62 aggregates[26]. Importantly, we also observed a reduction in the size of $RpL27A^{+/-}$ patches in male wing discs compared to female wing discs (Fig. 4h, i). These data, therefore, suggest that Minute cells are at a greater relative growth disadvantage when competing with wildtype cells in male wing discs relative to female. It is possible that these differences reflect cell autonomous differences in growth rates between male and female Minute cells, which manifest in clonal growth phenotypes in a competing, mosaic context.

## Quantitative analysis of the parameters of Minute competitive apoptosis using PECAn

Next, we exploited the sensitivity and high-throughput analysis capabilities afforded by PECAn to study the properties of cell competition-induced cell death in *Minute* cells. It is well established that competing *Minute* cells proximal to wild-type winners exhibit higher levels of cell death relative to distant competing *Minute* cells[14–16,27]. It is also well established that non-mosaic *Minute* cells exhibit elevated levels of cell-autonomous apoptosis[12–14,28]. However, how these levels of apoptosis – intrinsic vs competition-induced – compare has never been measured precisely. We therefore generated wing discs with a mosaic anterior compartment and an entirely *Minute* posterior compartment. As Minute cell competition does not occur across compartment boundaries[29], this allowed us to compare competitive and non-competitive cell death within a single tissue (Fig. 5). We then analyzed these samples in PECAn, utilizing its ability to assess multiple ROIs within a single image (Fig. 5b). Levels of death were greater in non-competing $RpS3^{+/-}$ cells relative to competing wildtype winners, confirming the previously reported observation that *Minute* cells

exhibit elevated levels of cell-autonomous apoptosis (Supplementary Fig. 6a)[14]. The density of apoptotic cells was higher at the competing *Minute* patch border relative to the internal reference death level observed in non-competing posterior compartment, indicating that competitive cell death represents an elevation over the non-competing condition (Fig. 6c). Furthermore, we observed no difference in levels of apoptosis between the non-competing posterior compartment and the patch centre in the competing anterior compartment, indicating that centre cells exhibit approximately baseline levels of $RpS3^{+/-}$ cell death (Supplementary Fig. 6b).

We then sought to use PECAn to rigorously identify the parameters of cell death within wing discs undergoing Minute cell competition. We compiled a large dataset of competing wing discs prepared using consistent conditions, but dissected, processed, and imaged in separate batches (67 wing discs and 192,207 $RpS3^{+/-}$ cells in competing conditions from six separate dissections). To control for the observed sexual dimorphism, only female larvae were dissected. Individual cells within all wing discs were assessed for their cell death (using PECAn single cell and foci segmentation) and spatial properties, and the resulting dataset was subjected to a multiple logistic regression-based analysis, using the PECAn statistical analysis app, with the dependent variable being whether or not a given cell is viable or apoptotic (Fig. 6a–f, Supplementary Data 2).

This analysis provides evidence for several interesting interactions within competing wing discs. We found that the probability of *Minute* cell death varies with the size of the wing pouch region (Fig. 6a, Supplementary Data 2). This could suggest that competitive death is more pronounced in slightly older discs. We also found that the rate of loser cell death declines as the number of $RpS3^{+/-}$ cells increases (Fig. 6b, Supplementary Data 2). As this analysis accounts for physical distance of $RpS3^{+/-}$ cells to the winner, this reduction in death is not exclusively due to a relative decrease in the exposure of loser cells to winners and could suggest that the strength of cell competition is sensitive to cell community effects influenced by the relative abundance of winners and losers, as has been reported for Rab5 and lgl-rasV12 cell competition[30,31]. Alternatively, the anticorrelation of loser cell death and loser cell abundance could simply reflect an underlying variability in the intensity of cell competition across wing discs, whereby weaker cell competition would both result in bigger $RpS3^{+/-}$ patches and in less competitive death.

Loser cell apoptosis also positively correlates with levels of apoptosis seen in winner cells (Fig. 6c, Supplementary Data 2), indicating that loser cells are more likely to undergo apoptosis in wing discs with higher overall levels of cell death. The probability of loser cell death further depends on a cell's position within the tissue, with basal locations associated with a higher probability of apoptosis (Fig. 6d, Supplementary Data 2). This result is consistent with established dynamics of cell death in wing discs generally and during Minute cell competition, specifically[16,32,33]. The probability of observing

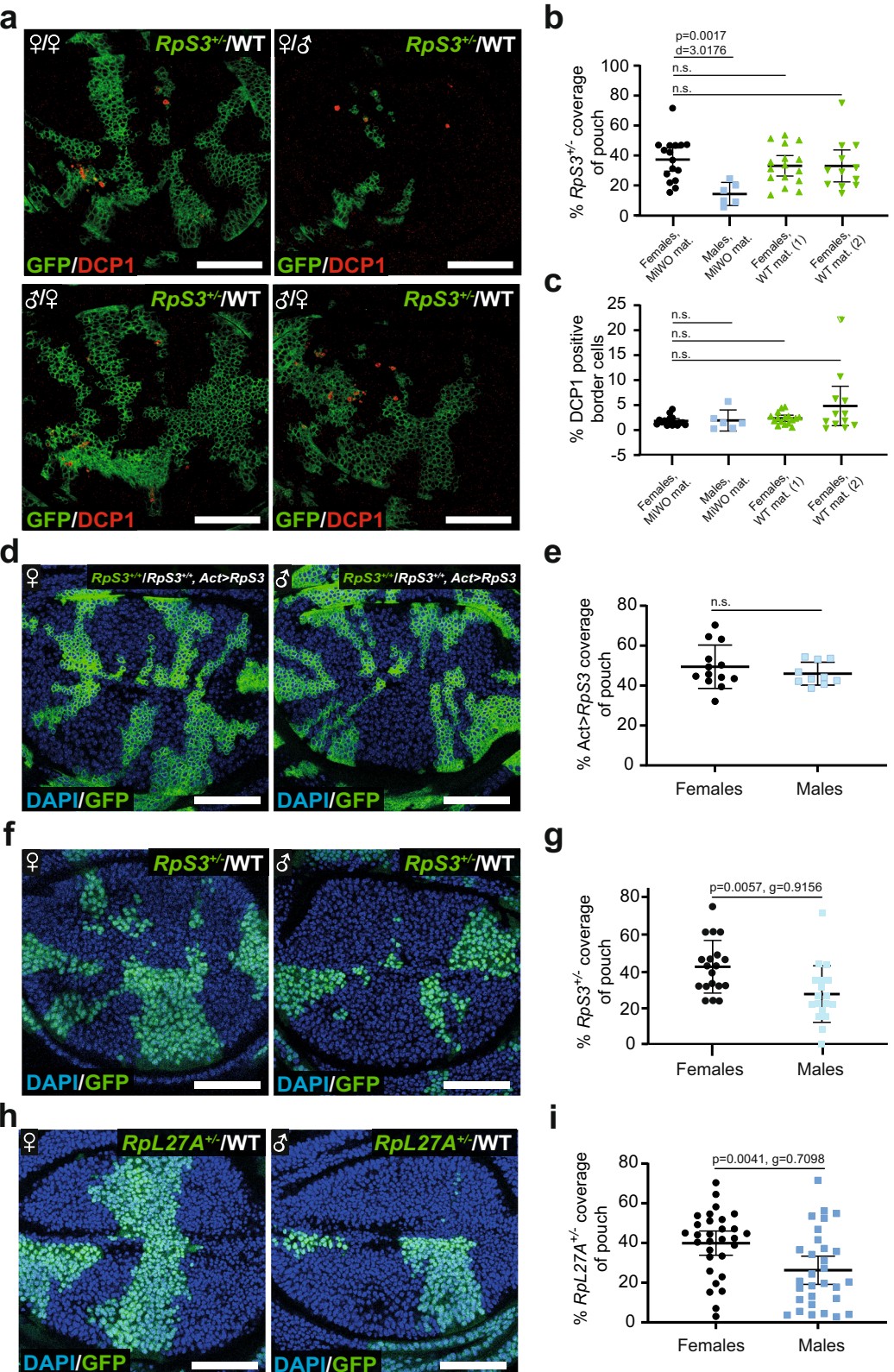

apoptosis also declines the further a loser cell is from the centre of the pouch (Fig. 6e, Supplementary Data 2).

Lastly, this analysis also identifies that loser cell death correlates strongly with physical proximity to winner cells. As multiple logistic regression analysis accounts for possible confounding factors in the dataset (e.g. patch size, shape, position, volume etc), it is significant that this analysis, while considering the relative

contributions of these factors, finds the probability of loser cell death increases exponentially the closer a cell is to wildtype neighbours (Fig. 6f, Supplementary Data 2). This relationship is consistent with previous analyses, which found that rates of loser cell death were highest within one-to-two cell diameters of the patch border[15,16]. It is also consistent with our data in Fig. 5. Altogether, these results confirm that the border death seen in

**Fig. 4 | Competing *RpS3*[+/−] losers exhibit a further growth disadvantage in male wing discs. a** Representative images of wing discs containing *RpS3*[+/−] losers (green) generated using the MiWO system competing against wildtype winners (unlabelled) and immuno-stained for cleaved-Dcp-1 (red). Wing discs were derived from female (top left) or male (top right) larvae (who inherited the rescuing construct from their mothers) or from two separate dissections of female larvae who inherited the rescuing construct from their fathers (bottom left and bottom right). **b** Quantification of *RpS3*[+/−] pouch coverage in wing discs as in (**a**), with mean and 95% CI shown. Statistics reflect two-sided *t*-tests without adjustment for multiple comparisons with un-pooled Cohen's d effect size. Biologically independent samples per replicate are as follows: replicate 1: $n_{Females, MiWO Mat.} = 16$, $n_{Males, MiWO Mat.} = 6$, $n_{Females, WT Mat. (1)} = 16$, $n_{Females, WT Mat. (2)} = 12$; replicate 2: $n_{Females, MiWO Mat.} = 14$, $n_{Males, MiWO Mat.} = 10$, $n_{Females, WT Mat.} = 8$; replicate 3: $n_{Females} = 18$, $n_{Males} = 15$. **c** Quantification of the percentage of cells undergoing apoptosis at the *RpS3*[+/−] patch border in wing discs as in (**a**), with mean and 95% CI shown. Statistics reflect two-sided Wilcoxon−Mann−Whitney *U*-test without adjustment for multiple comparisons. Biologically independent samples per replicate are as follows: replicate 1: $n_{Females, MiWO Mat.} = 16$, $n_{Males, MiWO Mat.} = 6$, $n_{Females, WT Mat. (1)} = 16$, $n_{Females, WT Mat. (2)} = 12$; replicate 2: $n_{Females, MiWO Mat.} = 14$, $n_{Males, MiWO Mat.} = 10$, $n_{Females, WT Mat.} = 8$; replicate 3: $n_{Females} = 18$, $n_{Males} = 15$. **d** Representative images of wing discs from female (left) or male (right) larvae containing *RpS3*[+/+] cells (green) in a background of unlabelled *RpS3*[+/+] cells carrying a third copy of the *RpS3* gene (*act > RpS3>Gal4*) and stained for DAPI (blue). **e** Quantification of *RpS3*[+/+] cell coverage in wing discs as in (**d**), with mean and 95% CI shown. Statistics reflect two-sided *t*-test with un-pooled Hedges g effect size. Biologically independent samples per replicate are as follows: replicate 1: $n_{Females} = 13$, $n_{males} = 10$; replicate 2: $n_{Females} = 19$, $n_{males} = 19$. **f** Representative images of wing discs from female (left) or male (right) larvae containing *RpS3*[+/−] losers (green) and wildtype winners (unlabelled) stained for DAPI (blue). **g** Quantification of *RpS3*[+/−] pouch coverage in wing discs as in (**f**), with mean and 95% CI shown. Statistics reflect two-sided *t*-test with un-pooled Hedges g effect size. Biologically independent samples per replicate are as follows: replicate 1: $n_{Females} = 15$, $n_{males} = 16$; replicate 2: $n_{Females} = 16$, $n_{males} = 15$; replicate 3: $n_{Females} = 19$, $n_{males} = 18$; replicate 4: $n_{Females} = 23$, $n_{males} = 20$. **h** Representative images of wing discs from female (left) or male (right) larvae containing *RpL27A*[+/−] losers (green) competing against wildtype winners (unlabelled) and stained for DAPI (blue). **i** Quantification of loser patch coverage in wing discs as in (**h**), with mean and 95% CI shown. Statistics reflect two-sided *t*-test with un-pooled Hedges g effect size. Biologically independent samples per replicate are as follows: replicate 1: $n_{Females} = 30$, $n_{males} = 30$; replicate 2: $n_{Females} = 21$, $n_{males} = 21$; replicate 3: $n_{Females} = 25$, $n_{males} = 31$. Scale bars correspond to 50 μm. Source data are provided as a Source Data file. ♀ symbol denotes females, ♂ denotes males, ♀/♂ denotes males who inherited the MiWO construct from their mothers.

competing wing discs is a fundamental feature of Minute cell competition.

## PECAn statistical regression analysis faithfully models and predicts competitive cell death

While this logistic regression model of loser cell apoptosis provides results consistent with the experimental literature, the model has a weak point which must be addressed: a Nagelkerke pseudo-$R^2$ value of 0.257. In the context of this particular model, the pseudo-$R^2$ can be thought of as a measure of how much better the model is at predicting which cells are apoptosing than a null model. This model, therefore, is poor at determining which cells will be apoptotic. There are two possible explanations for this observation: either there is a key predictor variable missing from this model, or the dependent variable has a high level of stochasticity. As the probability that a given cell will be apoptotic is low (1.4% of cells in the dataset are apoptotic), it seems likely that there is a strong element of stochasticity to exactly which cells are apoptotic at a given moment in time and that the model would struggle to accurately predict a rare, stochastic event. If the latter hypothesis is correct then the model should accurately predict the frequency of apoptotic cells at the level of the entire wing disc, as a population analysis would dilute the impact of stochasticity.

We therefore conducted a logistic regression analysis on the number of cells that are undergoing apoptosis versus the number that are non-apoptotic at competing Minute patch borders (Supplementary Data 3). The set of predictor variables used for this analysis was updated as shown in Supplementary Data 3, as some of the parameters used in the prior model were unsuited to this analysis. As this analysis was less computationally expensive, we could use an expanded dataset corresponding to 183 wing discs from 17 separate dissections. The resulting analysis predicts the number of apoptotic and non-apoptotic loser cells at *RpS3*[+/−] patch borders in each wing disc with a Nagelkerke pseudo-$R^2$ value of 0.9988 – indicative of an excellent fit between model and data. These results together indicate that, while rates of loser cell apoptosis increase according to several predictor variables, precisely which cells will undergo apoptosis is a stochastic process. The goodness-of-fit between this model and the data further indicates that this set of predictor variables provides a robust and comprehensive assessment of the parameters dictating levels of loser cell death at competing borders.

With this second analysis, we were furthermore able to evaluate how consistent metrics of cell competition are across experimental replicates, using both classical univariate and multiple logistic regression-based techniques (Fig. 6g, h, Supplementary Data 3). Replicates showed a degree of variability in the proportion of the pouch covered by *RpS3*[+/−] in these datasets (Fig. 6g). This is to be expected, as loser patches in this system are induced in a semi-stochastic fashion by heat-shock induction. These replicates, however, exhibited a high level of consistency at the level of border cell death, with only two of seventeen datasets – 3/9/19 and 9/8/20 - showing significant deviation from the base-mean, as determined via a Mann−Whitney U-test with an FDR p-adjustment and Cliff's δ effect size comparison (Fig. 6h). These data indicate that competitive loser cell death in Minute cell competition is a robust metric of cell competition that is refractory to noise across replicates.

## A PECAn-enabled RNAi screen identifies genes influencing minute cell competition

Having validated the accuracy and sensitivity of the PECAn pipeline in detecting and measuring Minute cell competition and having thoroughly tested MiWO as a tool for single cross induction of competing *Minute* cells expressing UAS-driven genes of interest, we sought to use them combined as a screening platform to identify genes involved in Minute cell competition.

We have previously shown that, even in the absence of cell competition, *RpS3*[+/−] cells and cells mutant in *Mahj*[−/−] (a functionally unrelated loser mutation) express a common signature of differentially expressed genes, relative to wild-type cells[17]. Many of these genes are predicted targets of *CncC*, the fly ortholog of the transcription factor Nrf2, which we have shown to be sufficient to induce the loser status[17]. This suggests that Nrf2 target genes expressed in *RpS3*[+/−] and *Mahj*[−/−] cells may modulate cell competition. Thus, we carried out RNA-seq of Nrf2 overexpressing wing disc cells and identified a list of putative Nrf2 targets (Supplementary Data 4). We then focused on the intersection of genes that are differentially expressed in all three loser inducing conditions (*RpS3*[+/−], *Mahjong*[−/−] and Nrf2 overexpression) and specifically at those genes that are upregulated, as Nrf2 is a transcriptional activator (Fig. 7a). This identified a list of 121 genes upregulated in all prospective loser conditions (Fig. 7a and Supplementary Data 5). We then ordered all available RNAi fly lines against those genes from VDRC's KK library (91 lines, corresponding to 87 genes) and used them to carry out a targeted RNAi screen to assess the impact of silencing those genes on Minute cell competition. Upon completion of the screen, we quantified all datasets created using PECAn. Each batch carried its own reference Minute cell competition control in which MiWO tool was crossed to the recommended control by VDRC corresponding

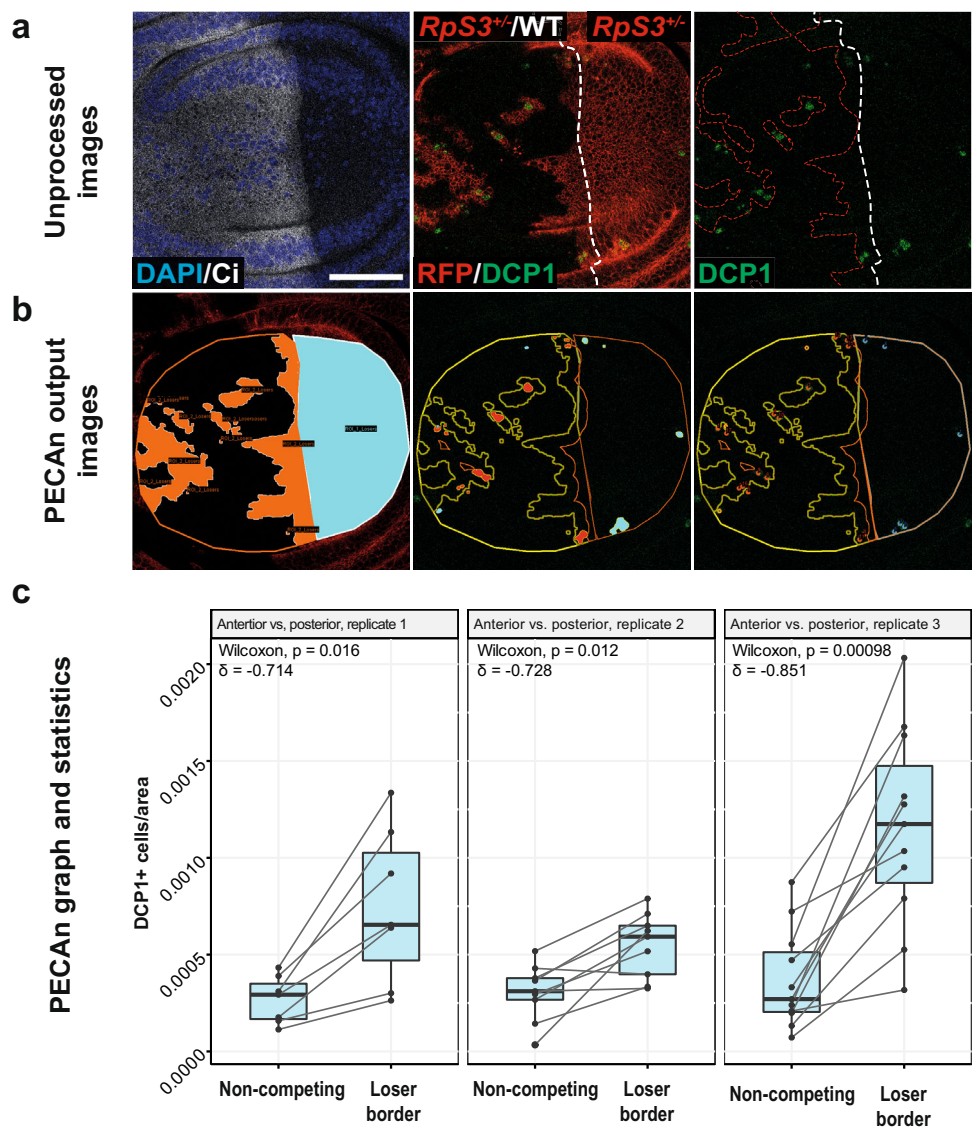

**Fig. 5 | *RpS3*⁺/⁻ cells exhibit higher levels of cell death during cell competition than in a non-mosaic context. a** Wing discs carrying a mosaic anterior compartment (identified by the anterior fate marker, Ci, shown in white) containing competing *RpS3*⁺/⁻ (red) and wildtype (unlabelled) cells and a posterior compartment (negative for Ci) that is entirely *RpS3*⁺/⁻. These samples were assessed for cell death via a staining for cleaved-Dcp-1 (green) and analyzed using PECAn. **b** PECAn output images identifying *Minute* cells in the anterior (orange) and posterior (cyan) compartments (left), the regions of the image positive for the Dcp-1 staining (middle) and the counts of individual Dcp-1-positive cells (right). **c** Output graphs and statistical tests generated by PECAn showing the density of Dcp-1-positive cells in the non-competing posterior compartment as compared to the *RpS3*⁺/⁻ patch border in the anterior compartment for three separate experimental replicates, wherein each dot corresponds to an individual wing disc. Box and whisker plot denotes minimum, first quartile, median, third quartile, and maximum. Statistics reflect 2-sided Wilcoxon signed rank test with Cliff's δ effect size metric. Biologically independent samples per replicate are as follows: replicate 1: *n* = 7; replicate 2: *n* = 9; replicate 3: *n* = 11. The scale bars correspond to 50 μm. Source data are provided as a Source Data file. Ci *Cubitus interruptus*, RFP = red fluorescent protein.

to the RNAi lines tested (see Methods). As a quality control step, all reference controls were pairwise compared to all other control groups for both *RpS3*⁺/⁻ pouch coverage and cell death metrics via two-tailed Wilcoxon–Mann–Whitney *U*-test. Those control groups which were scored as significantly different relative to the majority of other control groups were considered outliers, and these controls were discarded along with their corresponding batch. From the resulting 80 experiments (Supplementary Data 6), we identified 15 gene hits, which showed a statistically significant effect on *RpS3*⁺/⁻ pouch coverage and/or *RpS3*⁺/⁻ border death, as determined via a Mann–Whitney *U*-test with an FDR p-adjustment (Fig. 7b). Importantly, due to the use of automated image analyses afforded by PECAn, to the best of our knowledge, this is the first cell competition screen reported to use competitive death, in addition to patch size, as a screening parameter. This allowed us to identify RNAi

conditions that affect competitive death without necessarily affecting patch size (Fig. 7b), as is the case for the RNAi of the initiator Caspase, Dronc (Fig. 3h, i). One of the 15 hits was *Xrp1*, which has been previously implicated in Minute cell competition[18,19] and therefore acted as positive control. To further confirm the validity of the screen, we chose a subset of five of the hits (four that worsened cell competition metrics and one that rescued) to test reproducibility of the screen results: *Glutamate-cysteine ligase-catalytic subunit (Gclc)*, *Glutamate dehydrogenase (Gdh)*, *Abrupt (ab)*, *Vajk2*, and *Zormin*. For all of these genes, replication of these experiments confirmed the results from the screen (Fig. 7c), therefore confirming these RNAi lines as robust hits for modulators of Minute cell competition, the characterisation of which will be described elsewhere. Thus, the use of PECAn and of this screening methodology has yielded several promising cell competition leads.

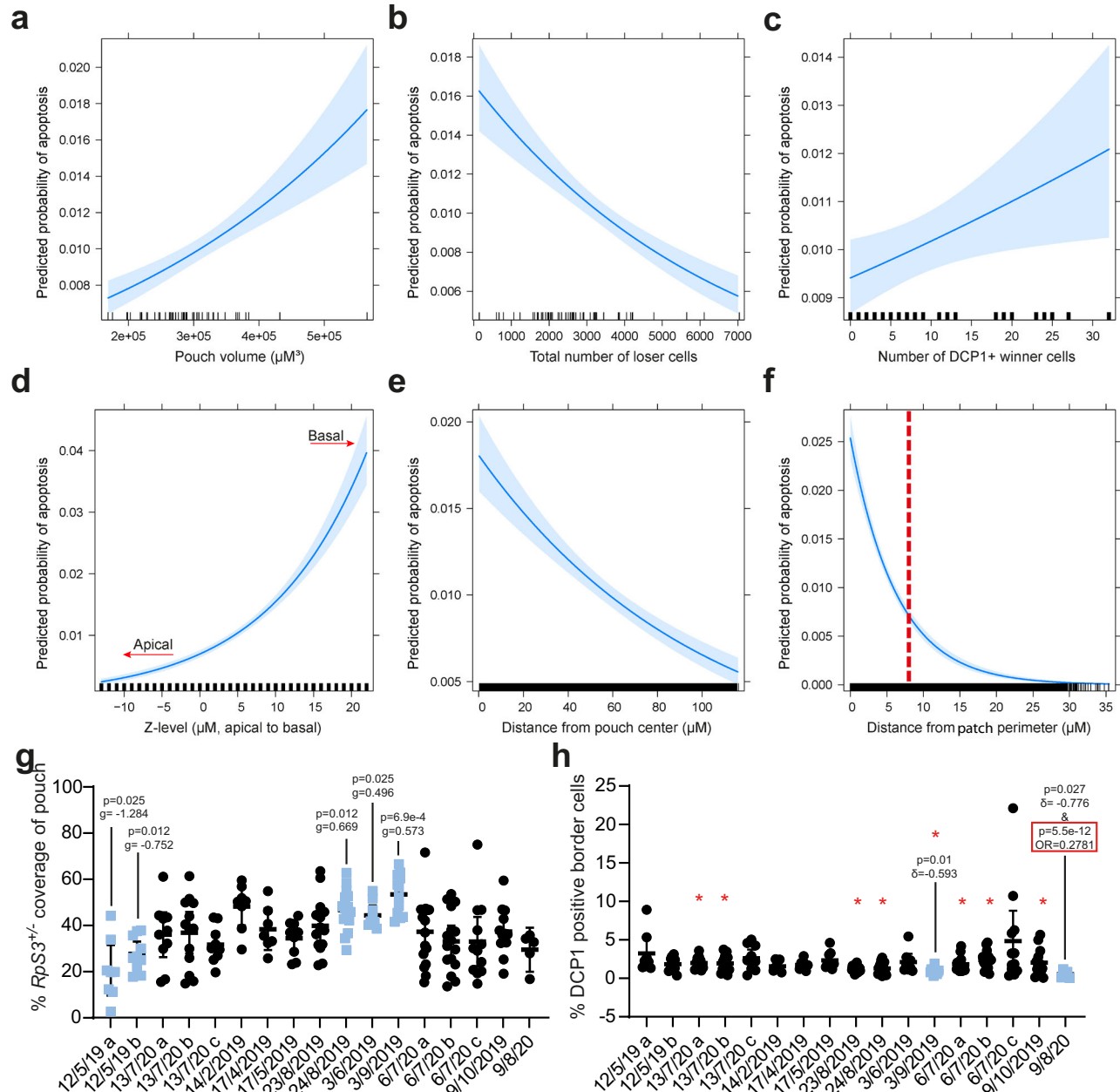

**Fig. 6 | Multiple logistic regression of competing wing discs reveals tissue parameters influencing loser cell death. a–f** Predicted effects plots of logistic regression analysis of individual cells in competing wing discs shown in Supplementary Data 2. In these, the probability of cell death is plotted against the pouch volume (**a**), the total number of $RpS3^{+/-}$ loser cells in the sample (**b**), the number of Dcp-1-positive wildtype winner cells in the sample (**c**), the Z-level of the cell (**d**), the distance of the cell from the centre of the pouch (**e**), and the distance of the cell from the loser patch perimeter (**f**). The plots in (**a–f**) show the expected probability of apoptosis upon changing a given predictor variable – the distance of the cell from the patch perimeter, whilst all other predictors are set to their mean value. Pale blue region denotes the 95% confidence interval, and black tick marks on the x-axis denote observed values in the dataset. The red dashed line in (**f**) is at 8 µm, which corresponds roughly to 2 cell diameters from the patch border and denoting the patch border/patch centre cut-off used in other analyses in this study.
**g, h** Comparison of replicates/experiments used in the logistic regression analysis shown in Supplementary Data 3. **g** The percent loser patch coverage of the wing

disc pouch region for each wing disc is shown on the y-axis, while the x-axis shows the replicate/experiment to which they belong, labelled by the date. p-values and g-values reflect two-sided student's t-tests with FDR correction and Hedges' g effect size metrics relative to the basemean. Measure of center and error bars are shown as mean and 95% CI, respectively. Biologically independent samples are, in order from left to right, n = 8, 10, 10, 13, 10, 8, 7, 9, 15, 15, 9, 12, 16, 16, 12, 8, 5. **h** The percentage of Dcp-1-positive loser cells is shown for each replicate/experiment. Statistics reflect either a two-sided Mann–Whitney U-test with FDR correction along with Cliff's δ effect size metric, relative to the basemean (no border) or the logistic regression analysis along with odds ratio effect size metrics (OR) shown in Supplementary Data 2 (red border). Measure of center and error bars are shown as mean and 95% CI, respectively. Biologically independent samples are, in order from left to right, n = 8, 10, 10, 13, 10, 8, 7, 9, 15, 15, 9, 12, 16, 16, 12, 8, 5. Datasets that were outliers by Mann–Whitney U-test are shown in blue, and red asterisks reflect datasets that had significant p-values by logistic regression but did not meet the effect size cut-off. Source data are provided as a Source Data file. OR odds ratio.

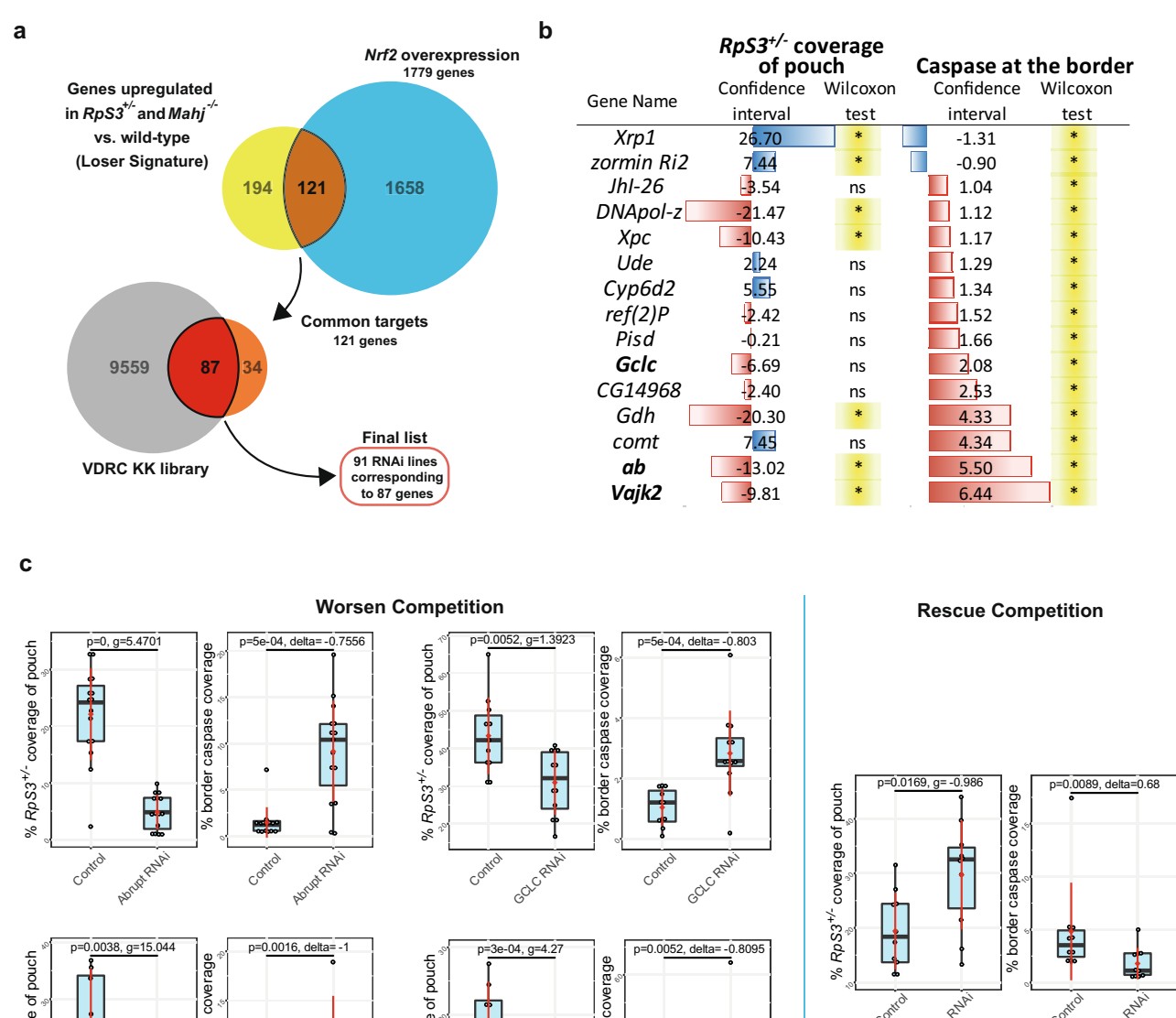

**Fig. 7 | A PECAn-powered RNAi-based screen identifies a list of cell competition genes. a** Schematic showing how target genes were identified. Initial hits were identified as those which were upregulated relative to the wildtype in wing discs heterozygous mutant for one of either of two loss-of-function alleles for *RpS3* and in homozygous *Mahjong* mutant wing discs. These initial hits were then refined by comparing them against genes upregulated in a third prospective loser condition, wing discs mildly overexpressing *Nrf2*. Of these 121 genes, we obtained 91 RNAi lines from the VDRC KK collection targeting 87 genes. **b** List of gene hits identified as candidate modulators of cell competition in the screen. MiWO wing discs expressing an RNAi were compared against control MiWO discs for both percent pouch coverage and percent cell death coverage at *RpS3⁺/⁻* patch borders. Hits were identified as those which yielded a statistically significant effect, as determined via a Wilcoxon–Mann–Whitney *U*-test with FDR p-adjustment, on either parameter.

**c** Post-screen validation of a subset of gene hits, showing replication of screen results for RNAi's targeting *ab, Gclc, Gdh,* or *Vajk2*, which exacerbate *RpS3⁺/⁻* cell competition and *Zormin*, which alleviates *RpS3⁺/⁻* cell competition. Statistics for the percent patch coverage reflect two-sided student's *t*-test with an un-pooled Hedges' g effect size metric, and statistics for the percent caspase coverage of the border reflect two-sided Wilcoxon–Mann–Whitney *U*-tests with a Cliff's δ effect size metric. Box and whisker plot denotes minimum, first quartile, median, third quartile, and maximum, whereas red diamond and whiskers denote mean and 95% CI. Biologically independent samples per condition are as follows: *Abrupt*: $n_{control} = 15$, $n_{RNAi} = 15$; *GCLC*: $n_{control} = 11$, $n_{RNAi} = 12$; *Gdh*: $n_{control} = 8$, $n_{RNAi} = 5$; *Vajk2*: $n_{control} = 9$, $n_{RNAi} = 7$; *Zormin*: $n_{control} = 14$, $n_{RNAi} = 10$ (replicate 1), $n_{control} = 10$, $n_{RNAi} = 10$ (replicate 2). Source data are provided as a Source Data file.

## Discussion

Clonal analysis techniques in situ provide unique and powerful methods for probing the behaviour of and interactions between heterogeneous cell populations. The challenges of extracting meaningful data from these samples, however, limits the speed, consistency, sensitivity, throughput and quality with which researchers can conduct these experiments, leading to under-analysed datasets, operator dependent variability/reproducibility issues, and low statistical power.

To address this, we have created a complete high-throughput image and data analysis pipeline for the 3D analyses of mosaic samples. This pathway is approachable and user-friendly even to biologists without programming or image analysis expertise. The underlying code is written in the powerful yet approachable Jython/Python and R languages, and researchers with programming experience will be able to readily incorporate other tools and algorithms into the pipeline. The software furthermore generates clear visual outputs of all measurements performed, allowing users to readily tailor the analyses to their samples.

The identification of a previously undescribed sexual dimorphism in minute cell growth in competing wing discs also highlights the utility of a high-throughput analysis approach. As the field generally has not accounted for gender before, this one factor might contribute to contrasting data seen in prior cell competition findings, and it may hamper sensitivity and discovery power. This is particularly relevant in screens for cell competition genes, where clone size is normally the parameter of choice[34–36], given that more labour-intensive analyses, such as for competitive death, cannot be carried out at scale, in the absence of an automated analysis tool like the one here presented. It remains unknown what mechanism accounts for this dimorphism: for instance, this difference in growth rates need not result from an explicit winner/loser interaction and could be due to differences in circulating levels of ligands. For instance, this difference could result from a sexual dimorphism in circulating levels of Unpaired ligands in *Drosophila*[22], signalling molecules which have been implicated in regulating both loser and winner cell growth and proliferation during competition[17,37]. It is not known whether the sexual dimorphism we describe reflects an effect on cell competition or whether it is due to autonomous differences in growth rates of *Minute* cells in males and females, which reveal themselves in a mosaic context. Irrespective of this, we suggest that studies of cell competition should account for gender in experiments evaluating for growth and clone coverage.

It is furthermore interesting that we observe an enrichment of p62 foci in *RpL27A*[+/−] loser cells, which contradicts a prior report[26]. This discrepancy may be due to the nature of the mutations used, as we used an allele (*RpL27A*[1]) different from the one (Df(2 L)M24F11) used in the previous study. As proteotoxic stress has been observed in *RpS3, RpS23, RpS26*, and *Mahj* mutants, a genetically unrelated though phenotypically similar mutation resulting in cell competition[13,14], our data showing that it is also found in *RpL27A* cells argue that proteotoxic stress is not exclusively a phenotype of small ribosomal subunit mutations.

A further advantage of this pipeline is that, with its built-in cell segmentation tools, it enables direct counts of apoptosing and non-apoptosing cells. Such an approach enables the use of more powerful and informative statistical techniques and analyses, such as those presented in this manuscript. These analyses have enabled us to investigate the parameters of competitive cell death with a high level of quantitative rigour.

The RNAi screen presented in this manuscript serves as a demonstration of the power of PECAn as a quantitatively rigorous and sensitive screening platform. It also provides a list of genes which merit further investigation, and which will likely lead to important advances in the mechanistic understanding of Minute cell competition. Importantly, some of the targets identified influence loser cell death without any corresponding effect on the size of loser patches. As prior cell competition screens have scored for conspicuous visible changes in clone size, these targets would likely not have been detected using prior screening methodologies.

Though the majority of these genes have not been previously implicated in Minute cell competition, many are involved in pathways which have previously linked to this process. For instance, *Pisd*[38] and *Gdh*[39] are genes with metabolic roles, and metabolic changes and signals have been shown to be essential in forms of cell competition

and supercompetition[40,41]. *Xpc*[42] and *Ude*[43] are involved in the DNA damage response, a process which has repeatedly been linked to cell competition[17,44,45]. The tools, analyses and gene datasets presented in this manuscript therefore will boost progress in our understanding of cell competition.

## Methods

### Ethical statement
We have complied with all ethical regulations for *Drosophila melanogaster* animal studies research, which are exempt from the Animals (Scientific Procedures) Act.

### Software development
The Pipeline for Enhanced Clonal Analysis (PECAn) software was designed in the script editor of ImageJ299 FIJI version 1.53d[9]. All code was written in the Jython 2.7.2 programming language. In addition to base FIJI packages, the Bio-Voxxel, MorphoLibJ[11], and the IJ-plugins toolkit were used. The R shiny analysis application was made in RStudio using R3.6.3 and shiny 1.5.0. The analysis app uses the following external libraries: PerformanceAnalytics, ggplot2, boot, MASS, car, RColorBrewer, ggpubr, markdown, ggsignif, rhandsontable, msm, dplyr, magrittr, ICSNP, mvnormtest, psych, corrplot, rcompanion, stringr, effsize, sandwich, ggthemes, shinyBS, reshape2, and effects. A list of statistical tests along with their associated R function is provided in Table 1. The software, source code, sample images, and instructional videos are provided in the supplementary electronic materials.

**Table 1 | Statistical packages incorporated into PECAn analysis tool**

| Test | Function | Package |
|---|---|---|
| Wilcoxon Signed Rank | compare_means | ggpubr v0.4.0 |
| Wilcoxon Rank Sum | compare_means | ggpubr v0.4.0 |
| Student's *t*-test | compare_means | ggpubr v0.4.0 |
| Paited *t*-test | compare_means | ggpubr v0.4.0 |
| Kruskal-Wallis | compare_means | ggpubr v0.4.0 |
| ANOVA | compare_means | ggpubr v0.4.0 |
| Shapiro-Wilks | shapiro.test | stats v3.6.2 |
| Fligner-Killeen | fligner.test | stats v3.6.2 |
| Fisher's exact | fisher.test | stats v3.6.2 |
| Cliff's delta | fliff.delta | effsize v0.8.1 |
| Cohen's d | cohen.d | effsize v0.8.1 |
| Hedge's | cohen d | effsize v0.8.1 |
| Spearman's rho | cor.test | stats v3.6.2 |
| Pearson's r | cor.test | stats v3.6.2 |
| Kendall's Tau | cor.test | stats v3.6.2 |
| P-adjustments | p.adjust | stats v3.6.2 |
| Correlation matrix | chart.Correlation | PerformanceAnalytics v2.0.4 |
| Multiple linear regression | Lm | stats v3.6.2 |
| Logistic regression | Glm | stats v3.6.2 |
| Poisson regression | Glm | stats v3.6.2 |
| Negative binomial regression | glm.nb | MASS |
| Variance inflation factors | Vif | car v3.0-10 |
| Durbin-Watson Test | durbinWatsonTest | car v3.0-10 |
| Non-constant variance of error | ncvTest | car v3.0-10 |
| Nagelkerke pseudo-$R^2$ | Nagelkerke | rcompanion v2.3.26 |
| Cox and Snell pseudo-$R^2$ | Nagelkerke | rcompanion v2.3.26 |
| McFadden pseudo-$R^2$ | Nagelkerke | rcompanion v2.3.26 |

## Fly husbandry and RpS3$^{+/-}$ cell induction

All flies were reared on wheat-based food prepared according to the following recipe: 7.5 g/L agar powder, 50 g/L baker's yeast, 55 g/L glucose, 35 g/L wheat flour, 2.5% nipagin, 0.4% propionic acid and 1.0% penicillin/streptomycin, and all experimental crosses were kept in an incubator at 25 °C. The parent cross was allowed to lay eggs for 24 h, and all dissections were performed on larvae at the wandering third instar stage, 6 days after the start of egg laying. RpS3$^{+/-}$ cell induction was accomplished using a heat-shock inducible FLP recombinase by placing fly vials in a water bath set to 37 °C. Specific heat shock timings and durations are listed in Supplementary Data 7. For RpS3$^{+/-}$ cells expressing temperature-sensitive constructs, fly vials were transferred to a water bath set to the specified temperature immediately following heat shock and were then dissected as normal. All conditions compared within an experiment were processed in parallel at the same time (e.g. eggs collected in the same 24 h window, crosses heat shocked at the same time and dissected at the same time). This includes comparisons between male and females, which were always taken from the same vial. Experimental genotypes, and RpS3$^{+/-}$ cell induction conditions are listed in Supplementary Data 7. Additional steps were taken to ensure consistency during performance of the screen. First, to minimise temperature fluctuations all experimental crosses were kept in a waterbath set to 25 °C. Furthermore, to control for genetic background and batch to batch variability, every batch had, as internal cell competition reference, a control cross where a relevant landing site control fly line from the KK RNAi collection, matched to the RNAi lines, was used. The control lines used were either the VDRC attP control line (cat#60100), when the RNAi line tested had an insertion only in the host strain attP site at the cytological location 30B, or the VDRC 40D-UAS control line (cat#60101), when the RNAi line tested was inserted in both host strain attP sites, at the cytological locations 30B and 40D.

The following fly stocks were used: RpS3$^{Plac92}$ (cat#BL5627) and RpL27A[1] (cat#BL5697), (Bloomington Drosophila Stock Centre), UAS-Xrp1-RNAi (cat#107860), attP control line (cat#60100), 40D-UAS control line (cat#60101), and UAS-Dronc-RNAi (cat# 100424) were obtained from the Vienna Drosophila Resource Centre. yw, UAS-myr-RFP, and hs-FLP;;FRT82B were provided by Daniel St. Johnston, hh-Gal4/TM6b was provided by Jean-Paul Vincent. w + /w-; tub > CD2>Gal4, UAS-CD8-GFP; tub-Gal80$^{ts}$ was provided by Bruce Edgar, and the tub»>>Gal4 driver was generated by flipping out this stock. UAS-Nrf2 was described in[46]. hs-FLP, UAS-CD8-GFP;; RpS3[Plac92], act > RpS3>Gal4/TM6b was described in[14]. Xrp1-LacZ was provided by Nicholas Baker[18]. All RNAi lines used in the screen are provided in Supplementary Data 6.

## RNA sequencing

RNA sequencing was performed as described in[17]. For the Nrf2 over-expression condition, larvae were of the following genotype: hs-FLP/+; tub»>>Gal4, UAS-CD8-GFP /+ ; UAS-nrf2 / tub-Gal80$^{ts}$. For the control condition, larvae were of the following genotype: hs-FLP / +; tub»>>Gal4, UAS-CD8-GFP /+; tub-Gal80$^{ts}$/+, Larvae were reared as normal but were maintained in a water bath set to 28 °C. L3 wandering females were selected.

## Dissection, fixation, and immunofluorescence

All larvae were washed once and then dissected at room temperature in phosphate-buffered saline (PBS). Hemi-larvae were then immediately transferred to pre-chilled PBS on ice and then fixed in 4% formaldehyde for 20 min at room temperature on a nutating mixer. Hemi-larvae were subsequently permeabilized for 20 min in room temperature 0.25% Triton X-100 in PBS (PBST) followed by a 30-min blocking step in 4% foetal bovine serum in PBST. Rabbit anti-DCP-1 antibody (Cell Signalling, cat#9578 S) was diluted 1:2500 in blocking buffer, and primary incubations occurred overnight at 4 °C on a rocker. Larvae were then washed three times for at least 10 min in PBST

at room temperature. Secondary antibody and DAPI were diluted in blocking buffer, and hemi-larvae were incubated in secondary antibody for a minimum of 45 min at room temperature, followed by 3X additional 10-min PBST washes and then mounted in VectaShield mounting medium and sealed with a coverslip and nail varnish. Rabbit anti-p-eIF2α (Cell Signalling, cat#3398 T) and mouse anti-beta galactosidase (Promega, cat#Z3784), mouse anti-beta galactosidase (Promega, cat#Z3781), were used at a concentration of 1:500 using the same protocol. Rat anti-Ci (DSHB #2A1) was used at a dilution of 1:1000, rabbit Rabbit anti-p62 (1:2,000) was provided by Tor Erik Rusten[47] and immunostaining was performed as in[14]. Secondary antibodies used were donkey anti-rabbit IgG Alexa Fluor 555 (1:500, Thermo Scientific, cat#A31572), donkey anti-rabbit IgG Alexa Fluor 488 (1:500, Thermo Scientific, cat#A21206), goat anti-rat IgG 647 (1:500, Thermo Scientific, A21247), goat anti-mouse IgG 555 (1:500, Thermo Scientific, A21127).

## Image acquisition

All slides were imaged on Leica SP-5 or SP-8 confocal microscopes using a 40x, 1.3 numerical aperture PL apochromatic oil immersion objective with Leica type F fluorescence immersion oil. LAS AF 2.7.3.9723 software was used for image acquisition. Wing discs were imaged at 1.4x digital zoom over the entire pouch region, with Z-steps corresponding to 1 μm, with the exception of the datasets in Fig. 4d–g and Supplementary Fig. 4, wherein only single Z-planes were acquired, as no evaluation of cell death was performed. All images were captured as 8-bit images at either 512×x512 or 1024×x1024 resolution using LAS X software.

## Time to pupariation assay

Flies from experimental crosses were allowed to lay eggs on normal fly food for 24 h. Tubes containing eggs were maintained in a 25 °C incubator. Emerging pupae were scored over time. Each genotypic condition was scored in 5-6 independent repeats. The number of pupae was normalised to the total amount of pupae per vial and plotted as the cumulative fraction of pupariating larvae per time. The data points were fitted with a sigmoid function using a nonlinear least squares method as in[17].

## Statistics and reproducibility

For univariate statistics, parametric assumptions were evaluated using a Shapiro-Wilks normality test and a Fligner-Kileen homogeneity of variance test. If parametric assumptions were satisfied, a two-tailed student's t-test was performed along with a Cohen's d or Hedge's g effect size. If assumptions were violated, a two-tailed Mann–Whitney U-test was performed along with a Cliff's δ effect size. A pairwise two-tailed Wilcoxon signed-rank test was performed on wing discs with a competing anterior and non-competing posterior compartment, as the datasets were paired, and assumptions were violated. Manual quantifications used for validation of the pipeline were made in FIJI 1.53d and statistical tests were run in Graphpad Prism 9.5.1 using the same workflow.

For the multiple logistic regression analysis shown in Fig. 6, the dependent variable was, for each RpS3$^{+/-}$ cell in all wing discs, whether or not the cell was undergoing apoptosis, as determined using combined single cell and foci segmentation in PECAn. Multiple logistic regressions were then performed in PECAn considering a range of non-multi-collinear terms (as determined by a variance inflation factor below 5) within the PECAn data outputs, and we filtered between alternate models by selecting for the lowest Akaike information criterion. Data transformations were considered but were not seen to noticeably improve the model. To correct for multiple comparisons, p-values were adjusted using the False Discovery Rate (FDR) technique. The same optimization was performed for the logistic regression analysis shown in Supplementary Data 3, however the dependent

variable was instead the number of apoptotic vs. non-apoptotic cells in the loser patch border for each wing disc. Source data for all quantifications and replicates is provided in the Supplementary Source Data File. For all experiments, no statistical method was used to pre-determine sample size. No data were excluded from the analyses, and researchers were not blinded to allocation during experiments and outcome assessments.

## Reporting summary

Further information on research design is available in the Nature Portfolio Reporting Summary linked to this article.

## Data availability

Source data are provided with this paper and are available as a source data file. The RNA-sequencing dataset generated in this study have been deposited in the Gene Expression Omnibus repository with the accession number of GSE181165, the SRA accession number is SRP330534. Source data are provided with this paper.

## Code availability

All code generated for this manuscript are publicly available and can be downloaded at GitHub at the following link[48] https://github.com/mebaumgartner/Michaels_Magic_Macro. The statistical analysis web app can be accessed at: https://michaelbaumgartner.shinyapps.io/Macro_Analysis_App/. The FIJI plugin can also be installed using the following update site: http://sites.imagej.net/PECAn/. Tutorial videos are available on Youtube: https://www.youtube.com/playlist?list=PLUx1yCRUR0JxCuzEyXeNh_YmceS7YfpCg.

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

## Acknowledgements
We wish to thank Rafael Carazo-Salas and his laboratory for their guidance in designing, building, and validating the image analysis pipeline. We also thank Stephen Cross and Anatole Chessel for their help in evaluating the software and Daniel Lawson and Susan Connolly for assisting us in implementing, interpreting, and understanding statistical packages in R. Thanks to Cristina Villa del Campo and Miguel Torres for sharing sample images of a postnatal mouse heart. Finally, we thank the Wolfson Bioimaging Facility for access to microscopes. This work was supported by a Cancer Research UK Programme Foundation Award to E.P. (Grant C38607/A26831) and Wellcome Trust Senior Research Fellowships to E.P. (205010/Z/, 16/Z and 224675/Z/21/Z). J.L. is supported by an EMBO Postdoctoral Fellowship (EMBO ALTF 947-2021).

## Author contributions
M.E.B. and E.P. led the project. M.E.B. designed, built, and tested the PECAn fiji plugin. M.E.B. and R.L. designed the R shiny app. M.E.B., P.F.L., R.L., J.L., A.M., A.N.T.i, and I.K. carried out experiments. M.E.B., P.F.L., R.L., A.M., and E.P. contributed to the experimental design. M.E.B., P.F.L. and E.P. wrote the manuscript with help from all other authors.

## Competing interests
The authors report no competing interests
