## [Peer Review File · Nature Communications]

Reviewers' Comments:

Reviewer #1:

Remarks to the Author:

This paper describes a new tool for measuring cell competition in 3D in *Drosophila* imaginal disc tissues. The authors have designed a pipeline for automated image processing and statistical analysis of mosaic tissues. The program is available as a plug-in for ImageJ/FIJI, taking advantage of that program's underlying algorithms. The program, which they call PECAN, seems to be quite useful for analyzing mosaic tissues as they describe. Following the tutorials supplied by the authors, I analysed their sample #1 for clones, using z-sections 1-13 to look at clone area, as described in the tutorials. I found the program to be a little difficult at first, but over time, with some repeats of the tutorials, I could follow and obtained images/results that matched the ones on the tutorial. I used the default mode with sample 1, and then tried PECAN to measure cell clones in mosaic tissues induced on my own laboratory. Although the measurement program is apparently programmable, for me at least, that will take some serious work.

The authors validate the program by analysing "clones" in the wing imaginal discs of *Drosophila*. The word "clone" is used loosely. In some cases - e.g. clones of WT cells in a Rp+/- background - what is being measured is marked Rp+/- cells that are not clonal, but surrounded by WT cell clones. Clone is a very specific term to describe cells within the same direct ancestral lineage. Ideally, "clone" should only be used to refer to a real clone, not to patches of cells that are non-clonal. As the methods used to generate clones are not all the same for each example ("flp-out" versus mitotic recombination), and as these differences and whether clones or patches are being measured is not made clear for the reader, readers who may be unfamiliar with the techniques could be confused. The authors also used PECAN to analyse data from a RNAi-based screen for modifiers of Minute cell competition, showing that it is useful for this purpose.

Overall PECAN seems to be a useful tool for analysing mosaic tissues, although those who are not computationally savvy might also find it difficult. As it stands now, it will be most useful for researchers who want to do the same type of analyses as the authors: ie Minute cell competition in the wing disc, a simple epithelium. Whether it will be useful for more complex tissues remains to be determined. More sophisticated researchers will no doubt be able to adapt PECAN to different settings by adding their own parameters, including other tissues, other organisms, and cell culture, as the authors suggest in Figure 1. However, several weaknesses in the paper, described below, suggest that this paper might be better suited for a more specialized developmental biology journal.

Major issues

1. For some experiments (described in Supp Fig. 3) the authors used the flp-out method to generate clones of RpS3+/- (RpS3[Plac92]) cells in wing discs expressing RpS3 cDNA under control of the actin promoter (act>RpS3>Gal4/+). Prior to flp-out excision of the RpS3 cassette, the RpS3+/- cells are rescued by the RpS3 cDNA. Post-flp-out, the cells are subject to competition by the surrounding act>RpS3>Gal4/+ cells. I need some clarification for these experiments: according to the Figure legend, the larvae were heat shocked at 3 days, which in WT, would be 72 hrs after egg laying. Given the size of the disc primordium at this point, numerous clones (potentially hundreds) would be generated. In a WT larva, the clones would then grow over the next 2 days, before the larvae begin to wander and undergo pupariation. In the RpS3[Plac92] mutant, which are severely delayed, the growth period would be longer. Is the rescue by the cDNA complete or partial? Does the flp-out at day 3 cause these larvae to slow development? Looking at the figure, the green component of the disc is quite large, so either the GFP-expressing clones have had a fairly substantial period of time to grow, or they consist of many clones that have fused. These are technical issues that may be only relevant to researchers working in *Drosophila*, but they are glossed over here and thus somewhat misleading. I realize that the authors' point here is the pipeline analysis, but this should not be at the expense of how the experiment was done. For those of us in the field, this information is critical for its interpretation. A good case in point is the analysis of cell death in clones: the prevalence of dying Rp+/- cells at the borders of WT clones has been documented many times, and is well established in the Minute cell competition literature at this point. As the authors know, this is in contrast to several other contexts of cell competition. However, I always wonder if the data for this are not completely unbiased. Often the

clones have been growing for several days and the GFP areas are fairly long and thin at this point, resulting in a near-border location for most Dcp1 cells in the Rp+/- territory. If the analysis was done earlier in the clonal growth period, when the WT cells have not taken over the majority of the disc, would one see more death in centers of the GFP+ cell patches or clones as well? The PECAN method seems well-suited to determine whether this is the case.

2. In addition, the authors examined cell death in and near RpS3+/- cells in an experiment in which clones of WT cells were generated in only half of the wing disc, using the posterior-specific driver HhGal4 to express UAS-Flp recombinase. The other half of the disc remained completely RpS3+/- . With an n=3 (with about 6 or fewer DCP1+ cells each), they conclude that there is more cell death in the competing half of the disc than in the non-competing half. This is an interesting experiment but to feel confident enough about the data for their stated conclusion, many more replicates and data points are needed, especially with the caveat noted above about clone/patch size and shape.

3. In the abstract the authors state that they "identified, by statistical regression analysis, tissue parameters that model and predict competitive death". This seems to be based on the location of the Dcp1+ cells. However, basal delamination is a common outcome after induction of apoptosis, but is not predictive of cell death, it is a consequence of the process of cell death.

Minor issues

Supp Table 4: (This is referred to in the text as Supp Table 7, please correct this). Please define "40D/+" in the relevant genotypes listed. In addition, the figures don't seem to correspond to Figure genotype list: Figure 4 does not have D, but Supp Table 4 lists a Figure 4D left and Figure 4D right; the genotype for Fig 5 is missing the FRT information; the list for Figure 6 is a little confusing, I assume it also applies to Supp Figure 2, and if so it would be appropriate to add that to the list. If not, please clarify. Also, in the text (line 550) this Table is referred to as Supp Table 7. Overall the supplementary files are confusingly labeled and often do not match what is cited in the text.

Lines 413-14: Data referred to here as in Supp Table 4, but this is confusing as noted above, since the genotype list is also named Supp Table 4 is list of putative NRF2 targets.

Line 424: In this screen, when put into PECAN, "we excluded all batches wherein the associated reference Minute cell competition control was an outlier relative to other batches, indicating that the data would likely be unreliable." What constituted an outlier? Also, this sentence is confusing, and makes the data seem somewhat arbitrarily selected.

Lines 544-547: Clone induction info. According to the genotype list in Supp Table 4, both the flip-out system and FRT-mediated mitotic recombination were used to generate clones. Were the same conditions for heat shock duration and timing used for both methods? The text here is confusing (the sentence is a bit incoherent). As mentioned above, it would be appreciated if more details were given about the different types of clone induction for individual experiments.

Lines 554-556: were the animals expressing the "control" KK RNAi line also scored with PECAN? Is that data included in the dataset? This would be a good control for the plugin as well.

Fig. 7: Was the RNAi effect compared in Dcp1+ cells at clone borders? And, are the "effect sizes" compared to M/+ clones without the RNAi? Please make these methods more clear in text and/or figure legend.

Typos throughout the manuscript need to be corrected.

Reviewer #2:

Remarks to the Author:

Nature Communications:

PECAn

This manuscript describes a pipeline for segmenting images of tissues consisting of mosaic genotypes with a primary application shown for evaluating cell competition in the wing. The pipeline was used to analyze Minute cell competition. Of interest, they found that there is sexual dimorphism in Minute Competition and identified new genes with a role in cell competition. The statistical analysis appears to be comprehensive and appropriate.

The pipeline could be very useful for other researchers in the field and new genes involved in cell competition were identified, but not characterized to a significant degree. New observations on cell competition phenotypes were reported and potentially interesting. However, many key issues were not really addressed in the manuscript satisfactorily.

A major concern is that the "3D" segmentation process was not illustrated in the figures (z-projected images appeared to be presented throughout). The 3D segmented volumes were never shown.

Second, the nuclear packing and potentially close apposition of nearby apoptotic cells is a clear consideration. It is often difficult to separate nuclei and to assign them to the correct cell (especially given the tightly packed architecture of the wing pouch). It would be very important for the authors to show that they can indeed correctly segment and identify the cell boundary and cytoplasm with its corresponding nucleus, especially near the border regions. These critical features are not really shown. In particular can all nuclei be correctly counted in a particular mosaic clone or is this only useful for identify sparse nuclear markers? As an example, p. 14, line 308-309 it sounds like cell numbers in mosaics can be quantified but this is not clear from data. This is not very trivial with wing discs.

High resolution close up images are not presented in the figures. Does the same tissue imaged at two different resolutions give similar results?

The authors have not developed a new algorithm but rather developed a tool utilizing a combination of established imaging approaches. This presents a concern of what is the major innovation or novelty of the work. The newly identified cell competition genes are presented, but little characterization was performed.

Computational imaging pipelines often prove themselves to be difficult to implement or reproduce by other groups. It is not clear that the pipeline will be able to be utilized by other groups. Some evidence that multiple users working from different microscopes and file formats successfully utilizing the tool should be presented.

It would be important for the screening results to attempt multiple RNAi lines for that gene and to rule out other artifacts by demonstrating that the knockdown was specific for that gene. Mere replication (repeating) the experiment is not sufficient

The Supplementary text (figures and tables) were poorly formatted and very difficult to comprehend or interpret.

Given these concerns, it is difficult to support publication of the manuscript.

p. 3 bespoke is an awkward word

p. 44 Figure 7 : typo "identifia"

Reviewer #3:

Remarks to the Author:

This paper presents a very timely and versatile novel image analysis and statistical toolkit for analysing mosaic tissue (or clones) in complex 3D tissues. It is very well designed into 2 steps: 1) FIJI based software for performing the clonal analysis in 3D and 2) an R Shiny application for statistical analysis and data plotting, etc. Both packages are importantly based on free open source software, and require no programming knowledge from the user, making the tool highly versatile for many biologists. They also include very thorough and well explained YouTube tutorials to explain how to use their framework.

Importantly, the authors prove the method's functionality and versatility by using it to discover some novel insights into cell competition, a phenomenon that requires detailed statistical clonal analysis methods. They showed that sexual dimorphism exists in cell competition and successfully used PECAn to analyse data from a genetic screen to find and confirm new gene candidates that affect cell competition.

Overall this is a very well designed toolkit that is highly suitable for publication in Nature Communications and will interest a wide audience. I only have one major recommendation, which is that I would like to see the authors supply their source code by using an open source platform such as Github. This will not only help its usage and visibility, but also provide future users with a platform to report issues and bugs.

Minor points:

- The introduction starts well, explaining concepts to a wide audience. However, I feel that a paragraph to compare this method with previous methods (even if they are limited) is needed. On the contrary, paragraph 3 (starting line 71) does not add much and could be removed.
- Segmentation works reasonably well, and you can add your own models via Weka. Is it possible to add your own images that are already segmented?
- Has a FIJI plugin description been created yet? <https://imagej.net/list-of-extensions>
- The FIJI GUI seems a bit intimidating. Is there a way to simplify them? I would recommend hiding some 'advanced' options and only display the 'important/basic' ones.
- Typo: Fig. 7 title 'identifia'.
- Line 414: ', We'

Best wishes,
Yanlan Mao

Reviewer #4:

Remarks to the Author:

The authors developed a new pipeline that will be a valuable tool for image processing and statistical data analysis of complex multi-genotype 3D images, especially for the cell competition community in both vertebrate and invertebrate models.

Subsequently, the authors applied this software to:

- a) Support that there is a gender difference in Minute competition,
- b) Perform statistical regression analysis and identify parameters that model and predict competitive cell death
- c) Analyze rigorously an RNAi-based screen to identify modulators of Minute cell competition.

Strength: This paper provides a useful tool for clonal analysis studies which has the potential to be highly appreciated in the future by many researchers and expedite significantly their studies.

Weakness: On the other side, this work does not significantly advance our knowledge of cell competition, due to the major points listed below.

Major points

- 1) One question that I have is how PECAn will discriminate (or better exclude) small clones (e.g. with clone diameter less than 4-5 cells), where the cells may die autonomously and have Dcp1 staining. In that case, will the program count them as cells dying via cell competition (since the whole clone will be within two cell diameter from the winners) or the program will exclude such clones? If the program excludes such clones, does this show a limitation of the software when cell competition is strong and loser clones are efficiently eliminated?
- 2) Sexual dimorphism in Minute competition is a really interesting hypothesis, which is weakly

supported by the existent data that the authors present. The genetic tool that the authors used to study Minute competition depends on many parameters that could be affected by gender (e.g. heat shock expression, perdurance of RpS3 protein/mRNA, FLP, larval growth differences). There are no control experiments ruling out some of these possibilities. Therefore the existent data do not provide the support that there are differences in Minute competition due to gender as they claim in both of their abstract (line 39) and in the title in the result section (line 272). To support their hypothesis, the authors would need at a minimum to perform some simple control experiments.

For example:

a) Repeat the same experiments by removing the RpS3[Plac92] element and check if there are gender differences in clonal sizes (genotype to be tested: hsFLP, UAS-CD8-GFP/+; act>RpS3>Gal4/+).

b) Repeat the same experiments by having hsFLP on other chromosomes

c) Repeat the same experiment in presence of Xrp1null mutation (e.g. Xrp1[M2-73] or Xrp1[08]) which has been shown by others that is sufficient to block competition.

d) Study Minute competition by using a different genetic tool and different Rp Minute (e.g. RpL14, RpS17).

3) The authors state that they identified modulators of Minute competition. For the modulators that their depletion by RNAi increases boundary cell death, it is not clear if they trigger a different cell competition mechanism which additively affects the cell competitiveness. In order to support that they modulate Minute competition, they should perform the same RNAi depletion in cells that do not have RpS3 mutation and see if and how competitiveness is affected.

4) Why the authors selected to confirm the validity of the screen by choosing only modulators that worsening the Minute competition (e.g. Gclc) and not novel modulators that rescue Minute competition (other than Xrp1, for example, zormin).

Minor points

1)The authors should mention the dissection time of the different genders. Do they observe differences in developmental delay between males and females that could explain the clonal size differences?

2)In line 316 they mention that the probability of Minute cell death varies with the size of the wing pouch region and that slightly older discs have more pronounced competitive death. Do the authors believe that PECan should somehow incorporate in its analysis the developmental delay of the dissected larvae or the size of the pouch? Maybe they have already incorporated this parameter in their analysis and I have missed it.

3)Line 261. Please also add PMID: 27574103 (2016), where it was first shown that Xrp1 heterozygous mutation protects Minute cells from cell competition

4)Line 324. Please add reference PMID: 20679206 (2010), where it was shown that merging of loser clones protects lgl- rasV12 cells (losers) from elimination by wild-type cells.

5)Line 484 "This confirms prior work". Please include citations.

6)Line 565: "Nrf2 overexpressing larvae were of the genotype: tub>Gal4, UAS-CD8-GFP / + ; UAS-nrf2 / tub::Gal80[ts]." Please provide details on the conditions used (time and temperature) to achieve the mild expression of Nrf2 overexpressing discs.

Reviewer #1 (Remarks to the Author):

This paper describes a new tool for measuring cell competition in 3D in *Drosophila* imaginal disc tissues. The authors have designed a pipeline for automated image processing and statistical analysis of mosaic tissues. The program is available as a plug-in for ImageJ/FIJI, taking advantage of that program's underlying algorithms. The program, which they call PECAN, seems to be quite useful for analyzing mosaic tissues as they describe. Following the tutorials supplied by the authors, I analysed their sample #1 for clones, using z-sections 1-13 to look at clone area, as described in the tutorials. I found the program to be a little difficult at first, but over time, with some repeats of the tutorials, I could follow and obtained images/results that matched the ones on the tutorial. I used the default mode with sample 1, and then tried PECAN to measure cell clones in mosaic tissues induced on my own laboratory. Although the measurement program is apparently programmable, for me at least, that will take some serious work.

We thoroughly appreciate the time and effort that this reviewer has invested in trialling directly the application we have generated. This is the best way to assess its validity, its power and its usefulness to the community. We are very happy to read that the reviewer was successfully able to measure cell clones in samples induced in their own laboratory. As we describe in the manuscript the application is off-the shelf ready to go without any need for further programming. We agree that although the application's capabilities do enable further programming for additional customisation, this requires additional effort.

The authors validate the program by analysing "clones" in the wing imaginal discs of *Drosophila*. The word "clone" is used loosely. In some cases - e.g. clones of WT cells in a Rp+/- background - what is being measured is marked Rp+/- cells that are not clonal, but surrounded by WT cell clones. Clone is a very specific term to describe cells within the same direct ancestral lineage. Ideally, "clone" should only be used to refer to a real clone, not to patches of cells that are non-clonal. As the methods used to generate clones are not all the same for each example ("flp-out" versus mitotic recombination), and as these differences and whether clones or patches are being measured is not made clear for the reader, readers who may be unfamiliar with the techniques could be confused. The authors also used PECAN to analyse data from a RNAi-based screen for modifiers of Minute cell competition, showing that it is useful for this purpose.

Thank you for pointing this out – a clarifying statement on the usage of the term 'clone' has been added to the manuscript (line 56).

Overall PECAN seems to be a useful tool for analysing mosaic tissues, although those who are not computationally savvy might also find it difficult. As it stands now, it will be most useful for researchers who want to do the same type of analyses as the authors: ie Minute cell competition in the wing disc, a simple epithelium. Whether it will be useful for more complex tissues remains to be determined. More sophisticated researchers will no doubt be able to adapt PECAN to different

settings by adding their own parameters, including other tissues, other organisms, and cell culture, as the authors suggest in Figure 1.

Given that the reviewer has only tested PECAN on clonal tissues similar to the ones thoroughly characterised in this manuscript, we appreciated that they may find it difficult to assess PECAN's general usefulness. While this software is most easily applied to experiments such as those used in the manuscript, it can also be applied to other 3D contexts and tissues. We have reported in the manuscript, as noted by the reviewer, that PECAN fares well in the 3D analysis of the *Drosophila* intestine and of the mouse heart (Figure 1). Furthermore, we are pleased to report that we have also obtained excellent feedback from several other investigators who have used PECAN for their analyses. For instance, the lab of Paola Bellosta (University of Trento, Italy, and New York University, USA) has enthusiastically used PECAN for their analysis of clonal and non-clonal experimental wing discs. Chris Bennett's lab, at the University of Pennsylvania, is using PECAN to analyze how well microglial transplants engraft in mice brains. The latter example shows that PECAN is already being used successfully by investigators, other than ourselves, in complex 3D tissues, such as the brain, and not just in simple epithelia.

However, several weaknesses in the paper, described below, suggest that this paper might be better suited for a more specialized developmental biology journal.

We have thoroughly addressed the weaknesses flagged by this reviewer in the revised manuscript, including several new experiments and modifications to PECAN. These are discussed in detail below.

Major issues

1. For some experiments (described in Supp Fig. 3) the authors used the flp-out method to generate clones of RpS3^{+/-} (RpS3[Plac92]) cells in wing discs expressing RpS3 cDNA under control of the actin promoter (act>RpS3>Gal4/+). Prior to flp-out excision of the RpS3 cassette, the RpS3^{+/-} cells are rescued by the RpS3 cDNA. Post-flp-out, the cells are subject to competition by the surrounding act>RpS3>Gal4/+ cells. I need some clarification for these experiments: according to the Figure legend, the larvae were heat shocked at 3 days, which in WT, would be 72 hrs after egg laying. Given the size of the disc primordium at this point, numerous clones (potentially hundreds) would be generated. In a WT larva, the clones would then grow over the next 2 days, before the larvae begin to wander and undergo pupariation. In the RpS3[Plac92] mutant, which are severely delayed, the growth period would be longer. Is the rescue by the cDNA complete or partial? Does the flp-out at day 3 cause these larvae to slow development? Looking at the figure, the green component of the disc is quite large, so either the GFP-expressing clones have had a fairly substantial period of time to grow, or they consist of many clones that have fused. These are technical issues that may be only relevant to researchers working in *Drosophila*, but they are glossed over here and thus somewhat misleading. I realize that the authors' point here is the pipeline analysis, but this should not be at the expense of how the experiment was done. For those of us in the field, this information is critical for its interpretation.

Thank you for your requests for further clarification. While our focus in this manuscript is indeed on the analysis pipeline, we have put a lot of emphasis and efforts on using the tool to generate novel and insightful data on Minute competition, from the analysis of death patterns in competing and non-competing contexts, to the discovery of sexual dimorphism in the behaviour of RpS3^{+/-} cells and the identification of novel modulators of cell competition. As such, we are keen to provide as much

clarity as possible and welcome your constructive requests to clarify, as they help us make sure our data can be appreciated by the community. Specifically, with respect to your questions:

- As indicated elsewhere above, and now explained also in the revised manuscript (line 56), we indeed use the term 'clone' to refer to patches of tissues of different genotype, without necessarily implying unique lineage. Under the experimental conditions that we use, it is indeed the case that some clones may have fused together. This however does not affect the validity of our results or their interpretation. Indeed, in all experiments we measure, and refer to, clone coverage of the pouch and not individual clone sizes. Differences in clone coverage can indeed be used to measure competition strength, as shown by our control experiments in Figure 3g and 3i. We further note that in some MiWO experiments, we purposefully induce large patches of green *RpS3^{+/-}* tissue, as this allows us to directly compare, within the same disc, cell intrinsic death levels (i.e. death observed in the center of those patches and away from competition) and competition-induced death levels (i.e. those levels observed at the periphery of the patches, where cells border WT cells).

- To address the questions regarding developmental timings of the various genotypes used, which we agree is important to control for, in the revised manuscript we provide a new figure analysing the time to pupariation for wildtype larvae, *RpS3^{+/-}* larvae, *RpS3* flp/out construct (MiWO) larvae without heat shock, and *RpS3* flp/out construct (MiWO) with heat shock (Supplementary Figure 3c). As we show, the MiWO construct substantially rescues the developmental delay of *RpS3^{+/-}* larvae, though the rescue is partial. Crucially, the data show that MiWO larvae have the same developmental timing, whether or not *RpS3^{+/-}* clones have been induced in them by heat-shock. This makes it a very convenient tool to use for competition studies, as that means that there are no differences in developmental timings that could act as confounding factor between homotypic MiWO and mosaic MiWO larvae.

A good case in point is the analysis of cell death in clones: the prevalence of dying Rp+/- cells at the borders of WT clones has been documented many times, and is well established in the Minute cell competition literature at this point. As the authors know, this is in contrast to several other contexts of cell competition. However, I always wonder if the data for this are not completely unbiased. Often the clones have been growing for several days and the GFP areas are fairly long and thin at this point, resulting in a near-border location for most Dcp1 cells in the Rp+/- territory. If the analysis was done earlier in the clonal growth period, when the WT cells have not taken over the majority of the disc, would one see more death in centers of the GFP+ cell patches or clones as well? The PECAN method seems well-suited to determine whether this is the case.

We agree with this reviewer that indeed PECAN's automated assignment of border vs center areas of a clone, alongside the MiWO genetic tool, can achieve unprecedented sensitivity and spatial resolution in determining the pattern of cell death and the extent/increase in cell death afforded by competing conditions. It is indeed with this goal that we have optimised experimental conditions, so as to be able to obtain sufficient data from both groups, border cells and center cells, and measure their relative death levels unequivocally. In this manuscript, we have done this in two complementary ways:

- 1) By generating large datasets of wing discs containing competing clones, we have analysed by single cell regression analysis close to 100,000 RpS3+/- cells from mosaic discs. Even though there are relatively fewer cells at the center than at the border, this still generates several thousands of center cells. Indeed, our single cell regression analysis specifically takes the distance of each cell relative to the border into consideration and shows a statistically measurable impact of distance from border on probability of cell death, as shown in Figure 6e and supplementary table 2.

2) We have generated an experimental genotype that allows us to compare, within individual wing disc, cell intrinsic death to competition-induced death in RpS3^{+/-} cells (Figure 5). Specifically, we have generated wing discs containing homotypic RpS3^{+/-} Posterior compartments, next to competing RpS3^{+/-} cells in Anterior compartments. As in this set up the entire A compartment is RpS3^{+/-}, the number of non-competing cells here is not limiting. Here too, we observe on a disc-by-disc basis a consistent increase in death as a consequence of cell competition (Figure 5c). While this set up is complementary to the one above, in that it does not allow a measurement of death as a function of the distance from clone borders, it does allow one to estimate with high precision the extent of cell death increase induced by cell competition, relative to death due to the RpS3^{+/-} mutation *per se*, as both are measured from within the same sample.

Together, we believe that the above data provide a conclusive demonstration, and the most accurate measurement to date, of competition-induced cell death at clone borders.

2. In addition, the authors examined cell death in and near RpS3^{+/-} cells in an experiment in which clones of WT cells were generated in only half of the wing disc, using the posterior-specific driver HhGal4 to express UAS-Flp recombinase. The other half of the disc remained completely RpS3^{+/-}. With an n=3 (with about 6 or fewer DCP1⁺ cells each), they conclude that there is more cell death in the competing half of the disc than in the non-competing half. This is an interesting experiment but to feel confident enough about the data for their stated conclusion, many more replicates and data points are needed, especially with the caveat noted above about clone/patch size and shape.

In this experiment, it is not three wing discs, it is three independent experiments. In the graphs, dots correspond to individual wing discs, not individual cells. We also would not have been satisfied with an n=3 wing discs with 6 cells each. Instead, the data is supported by three independent replicates, each of which contains multiple wing discs (n=6, n=9, n=8) wherein conditions are assessed within the same sample. To avoid confusion, a clarifying statement has been added to the figure legend.

3. In the abstract the authors state that they “identified, by statistical regression analysis, tissue parameters that model and predict competitive death”. This seems to be based on the location of the Dcp1⁺ cells. However, basal delamination is a common outcome after induction of apoptosis, but is not predictive of cell death, it is a consequence of the process of cell death.

We understand that the choice of wording may be confusing and have clarified it. This point of confusion is in reference to how the regression analysis functions, i.e. these parameters have predictive value for the regression analysis. Several tissue parameters and not just the position of a cell relative to the basal domain help the prediction. By analysing simultaneously parameters, such as disc size, position of cells from the clone border, and within the pouch, cell density, baseline death levels in the WT population, etc the analysis can accurately and quantitatively predict cell death probability. As basal delamination is often seen in dying cells it is only to be expected that the model finds this is as a useful predictive parameter. Indeed, if our analysis failed to pick up known features of cell death dynamics, we would have to question the analysis and why it failed to confirm these known findings.

Minor issues

Supp Table 4: (This is referred to in the text as Supp Table 7, please correct this). Please define “40D/+” in the relevant genotypes listed.

Thank you for pointing this out. 40D was used to refer to the cytologic band wherein the KK constructs were inserted. This has been updated with the specific genotype - P{attP,y[+],w[3]}. A description has also been included in the methods section at lines 445-449.

In addition, the figures don't seem to correspond to Figure genotype list: Figure 4 does not have D, but Supp Table 4 lists a Figure 4D left and Figure 4D right; the genotype for Fig 5 is missing the FRT information; the list for Figure 6 is a little confusing, I assume it also applies to Supp Figure 2, and if so it would be appropriate to add that to the list. If not, please clarify. Also, in the text (line 550) this Table is referred to as Supp Table 7. Overall the supplementary files are confusingly labeled and often do not match what is cited in the text.

Lines 413-14: Data referred to here as in Supp Table 4, but this is confusing as noted above, since the genotype list is also named Supp Table 4 is list of putative NRF2 targets.

Thank you very much for spotting those – We have made all necessary corrections.

Line 424: In this screen, when put into PECAN, “we excluded all batches wherein the associated reference Minute cell competition control was an outlier relative to other batches, indicating that the data would likely be unreliable.” What constituted an outlier? Also, this sentence is confusing, and makes the data seem somewhat arbitrarily selected.

Thank you for pointing this out, and a more detailed explanation of the methodology has been provided in lines 449-453. We aimed to be as consistent as possible with our experiments and to be mindful of outliers. Thus, each time we generated flies expressing an RNAi, we ensured that we included a control experiment. To ensure that the control experiment was valid – and thus that conclusions drawn from comparisons of an RNAi to that group – we discarded all experiments wherein the control experiment was an outlier. To find outliers, we created a matrix wherein we compared every control cross to every single other control cross via a pairwise, two-sided Wilcoxon test for both clone coverage and cell death metrics. The control groups from this matrix that were found to be a significant outlier relative to the majority of other control groups were deemed to be outliers, and thus those corresponding batches were excluded from further analysis.

Lines 544-547: Clone induction info. According to the genotype list in Supp Table 4, both the flip-out system and FRT-mediated mitotic recombination were used to generate clones. Were the same conditions for heat shock duration and timing used for both methods? The text here is confusing (the sentence is a bit incoherent). As mentioned above, it would be appreciated if more details were given about the different types of clone induction for individual experiments.

Thank you for pointing this out. This part of the text has been edited for clarity and detailed heat shock information for each individual experiment is provided in Suppl Table 7.

Lines 554-556: were the animals expressing the “control” KK RNAi line also scored with PECAN? Is that data included in the dataset? This would be a good control for the plugin as well.

Yes, they were. This has been clarified in the text.

Fig. 7: Was the RNAi effect compared in Dcp1+ cells at clone borders? And, are the “effect sizes” compared to M/+ clones without the RNAi? Please make these methods more clear in text and/or figure legend.

Thank you, this has been clarified in the figure legend

Typos throughout the manuscript need to be corrected.

Thank you for pointing this out

Reviewer #2 (Remarks to the Author):

Nature Communications:

PECAn

This manuscript describes a pipeline for segmenting images of tissues consisting of mosaic genotypes with a primary application shown for evaluating cell competition in the wing. The pipeline was used to analyze Minute cell competition. Of interest, they found that there is sexual dimorphism in Minute Competition and identified new genes with a role in cell competition. The statistical analysis appears to be comprehensive and appropriate.

The pipeline could be very useful for other researchers in the field and new genes involved in cell competition were identified, but not characterized to a significant degree.

We agree with this statement – however the purpose for including screen results in this paper was to establish that the pipeline could process and analyze large amounts of data with unprecedented sensitivity. While characterising the genes would fall beyond the scope of this manuscript, in the revision we have conducted further validation of the most prominent RNAi hits and confirmed that their modulatory effect on competition is reproducible. Further work on these and other screen hits will follow from our group. Equally, we hope that the community will dig into this gene list and use it as a resource to bring novel mechanistic insight into the process of cell competition.

New observations on cell competition phenotypes were reported and potentially interesting. However, many key issues were not really addressed in the manuscript satisfactorily. A major concern is that the “3D” segmentation process was not illustrated in the figures (z-projected images appeared to be presented throughout). The 3D segmented volumes were never shown.

Thank you for pointing this out – we have included a video showing the 3D analysis conducted by PECAn in various methodologies (Supplementary Video 1)

Second, the nuclear packing and potentially close apposition of nearby apoptotic cells is a clear consideration. It is often difficult to separate nuclei and to assign them to the correct cell (especially given the tightly packed architecture of the wing pouch). It would be very important for the authors to show that they can indeed correctly segment and identify the cell boundary and cytoplasm with its corresponding nucleus, especially near the border regions. These critical features are not really shown. In particular can all nuclei be correctly counted in a particular mosaic clone or is this only

useful for identify sparse nuclear markers? As an example, p. 14, line 308-309 it sounds like cell numbers in mosaics can be quantified but this is not clear from data. This is not very trivial with wing discs.

We were indeed able to accomplish this by combining membrane-tagged markers and nuclear stains. The macro performed as well as manual quantification by a trained researcher in identifying correctly single cells (Figure 3e-f).

High resolution close up images are not presented in the figures. Does the same tissue imaged at two different resolutions give similar results?

We have not investigated this point in detail, we have focussed our analysis on determining how well PECAN scores relative to manual annotation on images that were taken at the resolution that we have considered optimal for our experimental set up. We, as well as all the other investigators who have independently trialled PECAN with success (Uni Torino and U Penn for example; see response to Reviewer 1), have been able to apply it freely to their chosen image resolutions, which they had predetermined to be optimal for their experiments.

The authors have not developed a new algorithm but rather developed a tool utilizing a combination of established imaging approaches. This presents a concern of what is the major innovation or novelty of the work. The newly identified cell competition genes are presented, but little characterization was performed.

Even though these are established image-analysis techniques, they are currently not in an integrated an automated fashion for 3D analysis of complex tissues. Creating a formalized integration of techniques for comprehensive analysis, we feel, allows these powerful techniques to be usable in widely.

Computational imaging pipelines often prove themselves to be difficult to implement or reproduce by other groups. It is not clear that the pipeline will be able to be utilized by other groups. Some evidence that multiple users working from different microscopes and file formats successfully utilizing the tool should be presented.

This is an important point and we have sought further evidence of the wide applicability and usability of PECAN.

Regarding the compatibility with multiple file formatted, PECAN has been used successfully by us on .lif and .tif images and was also successfully used on .nd2 images provided to us by the Torres lab. Furthermore, the script is currently being used in Chris Bennett's lab at the University of Pennsylvania to assess mouse brains for engraftment by donor microglia. Images, microscope, and analysis are all done using separate instruments and operators from us.

Since initial submission of this work, we have distributed our pipeline to selected investigators, who have successfully implemented it to their images and tissue type. For instance, the lab of Paola Bellosta (University of Trento, Italy, and New York University, USA) has enthusiastically used PECAN for their analysis of clonal and non-clonal experimental wing discs. Chris Bennett's lab, at the University of Pennsylvania, is using PECAN to analyze how well microglial transplants engraft in mice brains. The latter example shows that PECAN is already being used successfully by investigators, other than ourselves, in complex 3D tissues, such as the brain, and not just in simple epithelia. We were also pleased to learn from Reviewer 1 that they were able to trial and successfully apply PECAN

it to their own images. It seems thus that there is little doubt that, once publicly available, PECAn will become a widely useful and accessible resource for the image analysis of complex 3D tissues.

It would be important for the screening results to attempt multiple RNAi lines for that gene and to rule out other artifacts by demonstrating that the knockdown was specific for that gene. Mere replication (repeating) the experiment is not sufficient

As indicated above, the purpose for including screen results in this paper was to establish that the pipeline could process and analyse large amounts of data with unprecedented sensitivity. While characterising the genes would fall beyond the scope of this manuscript, in the revision we have conducted further validation of the most prominent RNAi hits and confirmed that their modulatory effect on competition is reproducible. Further work on these and other screen hits will follow from our group. Equally, we hope that the community will dig into this gene list and use it as a resource to bring novel mechanistic insight into the process of cell competition.

The Supplementary text (figures and tables) were poorly formatted and very difficult to comprehend or interpret.

We have compiled supplementary figures and table according to accepted standards. however we are happy to make further modifications if required.

Given these concerns, it is difficult to support publication of the manuscript.

We hope that this thorough revision of the manuscript has allayed the above concerns.

p. 3 bespoke is an awkward word

This has been edited out

p. 44 Figure 7 : typo "identifia"

Thank you!

Reviewer #3 (Remarks to the Author):

This paper presents a very timely and versatile novel image analysis and statistical toolkit for analysing mosaic tissue (or clones) in complex 3D tissues. It is very well designed into 2 steps: 1) FIJI based software for performing the clonal analysis in 3D and 2) an R Shiny application for statistical analysis and data plotting, etc. Both packages are importantly based on free open source software, and require no programming knowledge from the user, making the tool highly versatile for many biologists. They also include very thorough and well explained YouTube tutorials to explain how to use their framework.

Importantly, the authors prove the method's functionality and versatility by using it to discover some novel insights into cell competition, a phenomenon that requires detailed statistical clonal analysis methods. They showed that sexual dimorphism exists in cell competition and successfully used PECAN to analyse data from a genetic screen to find and confirm new gene candidates that affect cell competition.

Overall this is a very well designed toolkit that is highly suitable for publication in Nature Communications and will interest a wide audience. I only have one major recommendation, which is that I would like to see the authors supply their source code by using an open source platform such as Github. This will not only help its usage and visibility, but also provide future users with a platform to report issues and bugs.

We thank this reviewer for their positive feedback and for recognising the value of the tool that we have generated.

We are happy to post these resources to such a platform – in fact we currently back the software up on Github privately and could simply make this public upon publication

Minor points:

- The introduction starts well, explaining concepts to a wide audience. However, I feel that a paragraph to compare this method with previous methods (even if they are limited) is needed. On the contrary, paragraph 3 (starting line 71) does not add much and could be removed.

Thank you for the suggestion. Paragraph 3 has been removed. As for a comparison with previous methods, this is the intended purpose of paragraph 2. The purpose of developing this tool is because there isn't really any integrated solution out there. There are no such public tools that that we are aware of that would perform a similar function.

- Segmentation works reasonably well, and you can add your own models via Weka. Is it possible to add your own images that are already segmented?

There is no explicit function for this, as people segment their images into any number of formats so it would not be straightforward to implement. However, one way to get around this is to use a weka classifier to segment the pre-segmented image – this would put the image into a consistent format.

- Has a FIJI plugin description been created yet? <https://imagej.net/list-of-extensions>

No – however we plan to do so upon publication

- The FIJI GUI seems a bit intimidating. Is there a way to simplify them? I would recommend hiding some 'advanced' options and only display the 'important/basic' ones.

Thank you for the excellent suggestions. We have indeed modified it and implemented a simpler user interface

- Typo: Fig. 7 title 'identifiea'.

Thank you, this has been fixed

- Line 414: ', We'

Thank you, this has been fixed

Best wishes,

Yanlan Mao

Reviewer #4 (Remarks to the Author):

The authors developed a new pipeline that will be a valuable tool for image processing and statistical data analysis of complex multi-genotype 3D images, especially for the cell competition community in both vertebrate and invertebrate models.

Subsequently, the authors applied this software to:

- a) Support that there is a gender difference in Minute competition,
- b) Perform statistical regression analysis and identify parameters that model and predict competitive cell death
- c) Analyze rigorously an RNAi-based screen to identify modulators of Minute cell competition.

Strength: This paper provides a useful tool for clonal analysis studies which has the potential to be highly appreciated in the future by many researchers and expedite significantly their studies.

Weakness: On the other side, this work does not significantly advance our knowledge of cell competition, due to the major points listed below.

Major points

1) One question that I have is how PECan will discriminate (or better exclude) small clones (e.g. with clone diameter less than 4-5 cells), where the cells may die autonomously and have Dcp1 staining. In that case, will the program count them as cells dying via cell competition (since the whole clone will be within two cell diameter from the winners) or the program will exclude such clones? If the program excludes such clones, does this show a limitation of the software when cell competition is strong and loser clones are efficiently eliminated?

PECan simply assigns death events as clones border events or clone center events, depending on proximity to WT cells and does not exclude clones on the basis of size. If the clone diameter is small the majority or entirety of death events will indeed be classed as border events. This is a valid concern, however it reflects a limitation of the clonal technique generation rather than a specific concern of the software. Given this issue, it is incumbent upon the experimenter to optimize clone induction conditions such that there is a sufficiently large amount of mutant cells proximal and distal to the winner cells. In our group, if a given experiment reveals increase in border death but insufficient resolution (owing to reduced clone size) to establish whether this is autonomous or not, we calibrate our heat-shock parameters to increase clone size.

2) Sexual dimorphism in Minute competition is a really interesting hypothesis, which is weakly supported by the existent data that the authors present. The genetic tool that the authors used to

study Minute competition depends on many parameters that could be affected by gender (e.g. heat shock expression, perdurance of RpS3 protein/mRNA, FLP, larval growth differences). There are no control experiments ruling out some of these possibilities. Therefore the existent data do not provide the support that there are differences in Minute competition due to gender as they claim in both of their abstract (line 39) and in the title in the result section (line 272). To support their hypothesis, the authors would need at a minimum to perform some simple control experiments.

We agree with this assessment and have therefore substantially reinforced these claims with new experiments, along the lines suggested by the reviewer below, which have been added in Figure 4d-g and supplementary figure 4.

For example:

a) Repeat the same experiments by removing the RpS3[Plac92] element and check if there are gender differences in clonal sizes (genotype to be tested: hsFLP, UAS-CD8-GFP/+;act>RpS3>Gal4/+).

This experiment has been completed and is now found in figure 4d-e. Consistent with the hypothesis of sexual dimorphism in *RpS3*^{+/-} competition, there is no observed difference between males and females in this context.

b) Repeat the same experiments by having hsFLP on other chromosomes

This experiment has been done, as included in supplementary figure 4a-b. Using a hs-FLP on the second chromosome, we in fact see a more pronounced sexual dimorphism.

c) Repeat the same experiment in presence of Xrp1null mutation (e.g. Xrp1[M2-73] or Xrp1[08]) which has been shown by others that is sufficient to block competition.

This experiment has been done and is now in supplementary figure 4c-d. A sexual dimorphism effect can still be observed even upon Xrp1 loss of function. This indicates that the pathway affecting dimorphic *RpS3*^{+/-} clonal growth is not dependent on Xrp1.

d) Study Minute competition by using a different genetic tool and different Rp Minute (e.g. RpL14, RpS17).

We have repeated the experiments inducing mitotic, FRT-based, *RpS3*^{+/-} clones. These too show sexual dimorphism (Figure 4f-g). This rules out that the effect is specific to the components that make the MiWO tool.

We have not investigated additional ribosomal mutations, as these further controls have already taken substantial time and effort, and given that this is not central to the main focus of the manuscript on PECAn. However, we have been careful throughout not to generalise the findings beyond *RpS3*^{+/-} and have further added a sentence in the discussion about this. Specifically we state: "Though these considerations will apply to experiments using RpS3+/-, a widely used mutation in the field, it remains to be seen if this sexual dimorphism phenotype is generalizable to Minute mutations other than RpS3+/-."

3) The authors state that they identified modulators of Minute competition. For the modulators that their depletion by RNAi increases boundary cell death, it is not clear if they trigger a different cell competition mechanism which additively affects the cell competitiveness. In order to support that they modulate Minute competition, they should perform the same RNAi depletion in cells that do not have RpS3 mutation and see if and how competitiveness is affected.

We agree that this is key to establish the mechanism behind the effect induced by specific RNAi depletion. This is indeed an experimental strategy which we perform regularly in the lab, however we feel that these experiments are beyond the scope of this particular manuscript. Here, we demonstrate that PECAn allowed us to establish the most sensitive cell competition screening pipeline to date, one where we not only score for clones size automatically, but where we can also automatically score for border death at scale. While characterising the genes would fall beyond the scope of this manuscript, in the revision we have conducted further validation of the most prominent RNAi hits and confirmed that their modulatory effect on competition is reproducible. Further work on these and other screen hits will follow from our group. Equally, we hope that the community will dig into this gene list and use it as a resource to bring novel mechanistic insight into the process of cell competition.

4) Why the authors selected to confirm the validity of the screen by choosing only modulators that worsening the Minute competition (e.g. Gclc) and not novel modulators that rescue Minute competition (other than Xrp1, for example, zornin).

This is a fair point. We now include validation of a total of 5 RNAi conditions, including zornin, as per reviewer's request (Figure 7c).

Minor points

1)The authors should mention the dissection time of the different genders. Do they observe differences in developmental delay between males and females that could explain the clonal size differences?

We have added a clarifying statement to the text at lines 547-550. For all gender experiments, males and females were collected from the same vial, thus the eggs were laid in the same 24 hour window, and they were heat shocked and dissected at the same time. While we observed no obvious developmental delay between males and females, this is not something that we specifically measured. However, in light of the additional male/female experiments which we have included in figure 4 and supplementary figure 4, we feel that a developmental delay could not account for the differences observed, as we have observed the same phenotype in various conditions wherein we were using different clone induction systems.

2)In line 316 they mention that the probability of Minute cell death varies with the size of the wing pouch region and that slightly older discs have more pronounced competitive death. Do the authors believe that PECAn should somehow incorporate in its analysis the developmental delay of the dissected larvae or the size of the pouch? Maybe they have already incorporated this parameter in their analysis and I have missed it.

This is indeed a feature of PECAn analysis. By performing regression analysis techniques, the size of the wing disc is indeed also considered as a variable and its ability to predict levels of cell death is considered. This allows us to gauge the effect of distance on cell death while also considering the relative contribution of other potentially confounding factors, such as size/age of the disc.

3)Line 261. Please also add PMID: 27574103 (2016), where it was first shown that Xrp1 heterozygous mutation protects Minute cells from cell competition

Thank you, this has been done.

4)Line 324. Please add reference PMID: 20679206 (2010), where it was shown that merging of loser clones protects Igl- rasV12 cells (losers) from elimination by wild-type cells.

Thank you, this has been done.

5)Line 484 “This confirms prior work”. Please include citations.

Thank you, this has been done.

6)Line 565: “Nrf2 overexpressing larvae were of the genotype: tub>Gal4, UAS-CD8-GFP / + ; UAS-nrf2 / tub::Gal80[ts].” Please provide details on the conditions used (time and temperature) to achieve the mild expression of Nrf2 overexpressing discs.

Thank you, this has been done.

Reviewers' Comments:

Reviewer #1:

Remarks to the Author:

The revised manuscript is definitely improved by the revisions made by the authors in response to the initial round of review. However, I still have the following concerns.

1. The inaccurate use of words tends to misrepresent the data that is presented. Although the long-standing techniques for mosaicism everyone uses in the field, if done with precision, will generate individual cell clones; however in this work the approach leads to large patches of merged clonal material, not clones themselves. Hence the the patches being measured are no longer "clones" (derived from the same mother cell), and should be referred to accurately. No need to reinvent words that already have specific and useful meanings.
2. The authors have attempted to address the above issue with a experiment in which one compartment is completely Rp-/+ , while the other compartment is a mosaic of wildtype and Rp-/+ cells. It is difficult to explain why they don't see cell death in the non-competing compartment, given the new findings (several labs including their own) about proteotoxicity in Rp-/+ cells even without competitive interactions. The literature suggests that these kinds of experiments leads to quite a bit of feedback between compartments, such that one may stop or slow its growth while the other is proliferating (e.g., Martin and Morata 2006, Wells et al 2006, Martin et al 2009, Vallejo et al 2015, Garelli et al 2015, Boulan et al 2018). The authors might consider the possibility that the P compartment in their experiments has stopped proliferating, which might have the effect of preventing/reducing the non-competitive cell death.
3. The use of the acronym "Minute in Wildtype Organism (MiWO)" is not accurate, since the RpS3 transgene in this context is under the control of the actin promoter, not its own promoter. Indeed, the authors contradict themselves since they state that the actin-RpS3 construct did not fully rescue the developmental delay of the RpS3-/+ mutant, clearly illustrating that it cannot be considered wildtype.
4. Anecdotal "data" from other labs who have tried PECAN: this is not data that can be analyzed in a review and should not be presented here as such.
5. This is a research paper, not a review, thus the text should be accurate and to the point. If written carefully and accurately, the data that illustrates the competence of their algorithm will speak for itself (obviating the need for sentences like that starting line 330).
6. Vague descriptions. For an analysis pipeline to be considered as a generally valid way to analyze cell competition, its design must be based on rigorously controlled experiments that are clearly described. The authors state (line 258) "In these wing discs, mutant cells carrying heterozygous mutations in the ribosomal small subunit protein RpS3 gene are generated in a mosaic fashion alongside wildtype cells and undergo Minute cell competition". This description is quite vague and will not be instructive to anyone new to or outside of the field of cell competition in *Drosophila*.
7. I disagree with the authors' argument that since the focus of the paper is the pipeline, the more definitive experiments re. the RNAis are beyond the scope of the work. They included the RNAi screen as part of PECAN validation, so the experiments should be rigorous and conclusive, or they should leave them out.
8. I tend to agree with Reviewer #2 that the major innovation /novelty of the work and algorithm is lacking. Various labs that work on cell competition have devised methods for scoring, which if explained carefully in each publication, are valid. The authors suggest that having uniformity of measurement would be an advantage to the field. However, one could argue that the many different scoring methods between labs, when done judiciously, will more accurately reflect the natural biological variation that each lab encounters (food quality, temperature, developmental synchrony methods, cell clone measurements vs patches normalized to areas of interest, etc). Results that are different between labs may well be due to real differences, that are worth exploring. A good example is raised by Reviewer #4 (major point #1) about PECAN's limitation in

scoring very small Dcp1 positive cells ("clones"?): are these due to the biological tendency of the Rp-/+ cells to die even with no competitive stress (as reported by the authors lab as well as several other labs recently)? Or are they due to cell competition? The wide spread of developmental timing of their larvae (+/- 24 hrs) also adds to the difficulty of these measurements. In their response, the authors make a point about "border events", but this assumes that "loser" cells always die at clone borders, which is not entirely clear and depends on the competitive context. By allowing the algorithm to "assume" that these cells are always "losers", the decision becomes biased. I completely agree that each experimenter is responsible for optimizing clone conditions; when writing up results, the authors are also responsible for making the descriptions accurate and clear.

9. The data describing the male-female differences is improved with the new experiments asked for by Reviewer 4. However, they find that Xrp1 is not required for the sexual dimorphic effect, suggesting that the differences they observe are agnostic to cell competition.

Reviewer #2:

Remarks to the Author:

The authors satisfied the critiques and questions raised in the initial submission. The most interesting aspects of the paper are the new quantitative insights gained about cell competition, including sexual dimorphism and the relative importance of stochasticity. If the authors incorporate the essential biological insights into the title it may alert more readers interested in factors impacting cell competition and growth and apoptosis -related phenotypes.

Reviewer #3:

Remarks to the Author:

All my concerns have been addressed. I therefore recommend the paper for publication. However I would like the authors to ensure code is shared (made public) on GitHub.

thank you,
Yanlan Mao

Reviewer #4:

Remarks to the Author:

I really appreciate the effort that the authors put into replying to my previous review comments (e.g. using different flippases, control clones, a different RpS3 competition genetic tool).

As I stated in my previous report the strength of this paper is that it provides a quite useful tool for clonal analysis studies with the potential to be highly appreciated by many researchers and expedite significantly their studies.

Weakness: On the other side, still, in my opinion, this work does not significantly advance our knowledge of the cell competition field, since their conclusion on the role of sexual dimorphism in cell competition is not strongly supported.

MAJOR POINTS

My main objection, still, is regarding the proposed role of sexual dimorphism in Minute cell competition.

The main reason is the result that gave one of the experiments that I proposed, which in my opinion does not agree with their conclusion. More specifically, Xrp1 is responsible for Minute cell competition, including RpS3 competition. This has been confirmed by many groups, including Dr Piddini's (30078730, 30531963,31841522, 33495633, 31909714, 34871307, 34914692,

35179490) and it is also shown in Figure 3h, where Xrp1-RNAi completely blocks competitive cell death (Dcp1 positive cells number is zero).

Therefore, since in Xrp1 mutant background the sex difference is still present (line 312-316, Suppl Fig 4c-f), my only interpretation is that this is independent of cell competition. Have the authors performed Dcp1 staining in RpS3-dependent competition in the absence of Xrp1, to check if Dcp1 is still present in males, which could explain the difference in sex?

It has been shown that Xrp1 substantially, but not completely, affects also the growth and the developmental delay of Minutes (PMID: 30078730). Therefore, my interpretation could have been that sex affects the growth of Minutes and not their competition (Meaning the growth of the female Minute cells is higher than males Minute).

On the other hand, it has been shown that by the end of larval development wing discs of wild-type females are larger than that of wild-type males (for example in Figure 6 of PMID: 28976974), supporting the increased larval growth in females that has been shown previously. So, I do not feel comfortable even stating a clear conclusion that the increased cellular growth in females is due to Minute mutation.

My concern is that maybe other parameters affect the sex difference in clonal size that is observed in Minute clones that are induced in wild-type tissue, but not in wild-type clones that are in wild-type tissue. Given the 24hr period of egg laying and the differential larval growth of males and females, for me, it is hard to draw straightforward conclusions.

In addition, having only one different Minute mutation, someone could argue that this has something to do with a specialized role of RpS3 protein and is not a general effect of Rp mutant cells. I indeed agree that they revised a significant part of the paper, but including a different Minute (for example RpL14) could have strengthened their conclusion, even regarding the role of sex in the growth of Minute cells.

Unfortunately, I can not agree that there is a substantial contribution in the cell competition field and my suggestion is that this work might be better suited in a more specialized journal.

Best regards.

RESPONSE TO REVIEWER COMMENTS

Reviewer #1 (Remarks to the Author):

The revised manuscript is definitely improved by the revisions made by the authors in response to the initial round of review. However, I still have the following concerns.

We thank the reviewer for acknowledging that our revisions have improved the manuscript, and we thank them for the time and effort invested in providing valuable criticisms. We have addressed the outstanding criticisms in the manuscript and in the point-by-point rebuttal below.

1. The inaccurate use of words tends to misrepresent the data that is presented. Although the long-standing techniques for mosaicism everyone uses in the field, if done with precision, will generate individual cell clones; however in this work the approach leads to large patches of merged clonal material, not clones themselves. Hence the the patches being measured are no longer “clones” (derived from the same mother cell), and should be referred to accurately. No need to reinvent words that already have specific and useful meanings.

We have altered our terminology to refer to these merged regions as ‘patches’ rather than clones.

2. The authors have attempted to address the above issue with a experiment in which one compartment is completely Rp-/+ , while the other compartment is a mosaic of wildtype and Rp-/+ cells.

The experimental approach the reviewer refers to was set up to have a very sensitive metric to compare autonomous phenotypes vs non-cell autonomous, competition-induced phenotypes, in *RpS3^{+/+}* cells. These can be directly compared within the same wing disc, making this set up exquisitely sensitive.

It is difficult to explain why they don’t see cell death in the non-competing compartment, given the new findings (several labs including their own) about proteotoxicity in Rp-/+ cells even without competitive interactions. The literature suggests that these kinds of experiments leads to quite a bit of feedback between compartments, such that one may stop or slow its growth while the other is proliferating (e.g., Martin and Morata 2006, Wells et al 2006, Martin et al 2009, Vallejo et al 2015, Garelli et al 2015, Boulan et al 2018). The authors might consider the possibility that the P compartment in their experiments has stopped proliferating, which might have the effect of preventing/reducing the non-competitive cell death.

This is an interesting suggestion, and we thank the reviewer for bringing it up. We have considered it and addressed in the manuscript and below. The conclusion is that our data, including new analysis we have added to address this, rule out this possibility.

1) First, we do see substantive levels of death in the anterior compartment – it is just that competing loser cells exhibit an increase in cell death above these baseline levels. Indeed, an additional analysis of our samples shows increased cell death in the non-competing posterior compartment relative to competing wild-type cells in the anterior compartment (Supplementary figure 5a). Specifically, analysis

of non-competing A compartments vs competing wild-type cells in the P compartments of the same wing discs shows increased death in the non-competing *Minute* territories, relative to wildtype.

2) In further analysis, we see that there is no significant difference in levels of cell death between the non-competing A compartments and the *Minute* cells in center-region cells in the competing compartment B, that is, *Minute* cells that are protected from competition (Supplementary figure 5b). This rules out the suggestion that A cells are protected from cell death by developmental stage or other.

The reviewer may be thinking of a rescue because of the reports of high autonomous death in *Minute* cells from other labs, but it is well established that such phenotypes are influenced by diet/nutritional content and (our own data indicates) even temperature. The bottom line is that, in the rearing conditions that we use for our studies, it is readily possible to appreciate both a higher autonomous death level in *Minute*/relative to wild-type cells and a substantially higher frequency of apoptosis during competition. Data from the Baker lab (Li and Baker, Cell 2007; Baker, Nature Rev. Genetics 2020) are also consistent with this.

3. The use of the acronym “Minute in Wildtype Organism (MiWO)” is not accurate, since the RpS3 transgene in this context is under the control of the actin promoter, not its own promoter. Indeed, the authors contradict themselves since they state that the actin-RpS3 construct did not fully rescue the developmental delay of the RpS3-/+ mutant, clearly illustrating that it cannot be considered wildtype.

We have amended the name of the construct to ‘Minute in Wildtype-like Organism.’

4. Anecdotal “data” from other labs who have tried PECAN: this is not data that can be analyzed in a review and should not be presented here as such.

As requested, we take back our prior comments. It will only be possible to evaluate how broadly applicable this tool is, once it is published and publicly available, as is the case for the great majority of tools/technologies/discoveries, the impact of which can only be assessed after publication. We do note, however, that Reviewers 2, 3, and 4 are all very positive about PECAN and its projected impact to the community.

5. This is a research paper, not a review, thus the text should be accurate and to the point. If written carefully and accurately, the data that illustrates the competence of their algorithm will speak for itself (obviating the need for sentences like that starting line 330).

We thank the reviewer for this criticism. We have revised the text throughout accordingly. With regards to the specific sentence cited, there is no sentence that we can see starting at line 330, so we are unable to establish what specifically needed changing there.

6. Vague descriptions. For an analysis pipeline to be considered as a generally valid way to analyze cell competition, its design must be based on rigorously controlled experiments that are clearly described. The authors state (line 258) “In these wing discs, mutant cells carrying heterozygous mutations in the ribosomal small subunit protein RpS3 gene are in a mosaic fashion alongside wildtype cells and undergo *Minute* cell competition”. This description is quite vague and will not be instructive to anyone new to or outside of the field of cell competition in *Drosophila*.

Thank you. We have edited the text to provide a more informative description.

7. I disagree with the authors' argument that since the focus of the paper is the pipeline, the more definitive experiments re. the RNAis are beyond the scope of the work. They included the RNAi screen as part of PECAn validation, so the experiments should be rigorous and conclusive, or they should leave them out.

Our experiments are indeed rigorous, they simply do not extend to include a characterization of the cell competition hits identified. They do however, thanks to the detailed phenotypic analysis afforded by PECAn, quantitatively score and separate the component of tissue coverage from that of competitive death for each identified hit, which is more than most reported screens in our field do. As the reviewer probably knows, it is standard in the competition field to report hits of screens without added follow up, leaving the characterization to future papers (Rhiner et al., Dev Cell 2010). In some instances, specific mutations are reported before the gene affected is identified (Tyler et al, Genetics 2007), thus this manuscript is in keeping with the standards of the field.

In addition, this manuscript does not simply report the screen. In this manuscript we have: developed PECAn, a state of the art 3D image and statistical analysis platform; tested and benchmarked it against manual quantifications (the standard in the field), showing that it perform at least as well, but without the labour and the operator dependent variability involved in human analysis; reported and characterized a sexual dimorphism component to Minute competitive growth; carried out a high-content data and statistical regression analysis of the features of competitive death; and performed an RNAi screen of nearly 100 genes upregulated in prospective *RpS3*, *NRF2OE* and *mahj* losers, releasing the list of gene hits.

Considering all the above, we believe it is fair to suggest that further analysis of the screen hits is warranted, but better served by follow up work.

8. I tend to agree with Reviewer #2 that the major innovation /novelty of the work and algorithm is lacking.

Reviewer #2 has now recommended the revised manuscript for publication. They state: "The authors satisfied the critiques and questions raised in the initial submission. The most interesting aspects of the paper are the new quantitative insights gained about cell competition, including sexual dimorphism and the relative importance of stochasticity." We would like to add that none of this would have been possible without PECAn, which has enabled sensitive and quantitative analyses at scale. To our knowledge there are no reported, publicly available computational tools for complex clonal 3D analysis, making PECAn a valuable tool for 3D tissue biologists.

Various labs that work on cell competition have devised methods for scoring, which if explained carefully in each publication, are valid.

Up until recently, our lab relied on manual quantifications, using Fiji or other standard image handling software. To our knowledge, this is true for most labs in the field. While these are indeed valid, the tools provided here enable more sensitive and diverse quantifications and they do so at scale, empowering researchers to carry out superior data analyses. The novelty is primarily two-fold: first, we provide a reliable automation of the scoring metrics that are already commonly in use, providing an important

tool to facilitate research in the field. Secondly, this platform enables several new, sophisticated measurements that have not been feasible beforehand, such as the single cell measurements with regression analysis.

The authors suggest that having uniformity of measurement would be an advantage to the field. However, one could argue that the many different scoring methods between labs, when done judiciously, will more accurately reflect the natural biological variation that each lab encounters (food quality, temperature, developmental synchrony methods, cell clone measurements vs patches normalized to areas of interest, etc). Results that are different between labs may well be due to real differences, that are worth exploring.

We agree that differences between labs can represent real biology (see for example response to point 2 above) and are worth exploring. We do not advocate for uniformity of measurements, and, like this reviewer, we believe that choosing different scoring parameters is important to extract new biology, much like adjusting experimental design is. In fact, on the contrary, PECAN rather than imposing standard parameters, is a tool that empowers the researchers to be creative about the parameters they wish to score, as myriad different measurements are possible. PECAN is a discovery and enabling tool. At the same time, though, the automated features make its measurements robust to changes in operator, effectively removing the error introduced by technical noise. As any scientist knows, controlling for technical error is essential to capture biological variability across samples. PECAN provides a tool to minimize technical error while increasing the sensitivity to capture biological differences, according to the parameters chosen by each investigator.

A good example is raised by Reviewer #4 (major point #1) about PECAN's limitation in scoring very small Dcp1 positive cells ("clones"?): are these due to the biological tendency of the Rp-/+ cells to die even with no competitive stress (as reported by the authors lab as well as several other labs recently)? Or are they due to cell competition? The wide spread of developmental timing of their larvae (+/- 24 hrs) also adds to the difficulty of these measurements. In their response, the authors make a point about "border events", but this assumes that "loser" cells always die at clone borders, which is not entirely clear and depends on the competitive context. By allowing the algorithm to "assume" that these cells are always "losers", the decision becomes biased. I completely agree that each experimenter is responsible for optimizing clone conditions; when writing up results, the authors are also responsible for making the descriptions accurate and clear.

Here two aspects of the experimental set up are raised: experimental design (PECAN independent) and data analysis (PECAN dependent). We discuss and clarify them separately below. Accordingly, we have further explained those points in the manuscript, to enhance clarity.

- Data analysis: PECAN is competition agnostic, it simply assigns death events as border events or center events, depending on proximity to WT cells; it does not label cells as losers and it does not exclude clones on the basis of size. There are therefore no biases or assumptions introduced by PECAN during the analyses. The quantification, without any biases included, scored increased border death relative to center death, which has in fact been reported several times before (Li and Baker cell 2007, Baker Nature reviews, 2020, Kucinski et al., 2017, Baumgartner et al., NCB 2021) and which we analyse with more precise tools in this manuscript. The single cell analysis

presented in Figure 6 makes even fewer assumptions. In that instance, we assess every cell for whether or not it is undergoing death and compute the distance from the winner/loser interface to evaluate its predictive power. In that context, we again see a dramatic increase in the probability of cell death at the loser border.

- Experimental design: as the reviewer rightly points out, clones that are too small are not suitable to distinguish between autonomous and competition induced death. Optimizing clone induction conditions, such that there is a sufficiently large number of mutant cells far from winner cells overcomes this problem. This allows direct comparison of border to center death within the same discs, providing a very sensitive, internally controlled measurement of competitive death. Through this manuscript, we have optimized clone induction conditions to favour the formation of large patches of loser territory over the ability to induce individual (not fused) clones to obtain large enough territories to compare intrinsic versus non-intrinsic (competition induced cell death).

9. The data describing the male-female differences is improved with the new experiments asked for by Reviewer 4. However, they find that *Xrp1* is not required for the sexual dimorphic effect, suggesting that the differences they observe are agnostic to cell competition.

Our carefully controlled experiments, including the *Xrp1* experiment the reviewer refers to, allow us to conclude that the competitive growth of *RpS3^{+/-}* cells is sexually dimorphic. Dimorphic competitive growth, in turn, impacts on the cell competition outcome in a largely *Xrp1*-independent manner.

The argument that because a phenomenon does not require *Xrp1* it cannot impact or result from cell competition is not necessarily correct. For example, any factor that modulates competition downstream of *Xrp1* will modulate cell competition in a *Xrp1*-independent manner. In addition, it cannot be ruled out that there are additional pathways that modulate cell competition that act in parallel rather than in the same pathway as *Xrp1*. These too would be predicted to modulate cell competition in a *Xrp1*-independent manner. Our reported role of JAK/STAT in the competitive growth of wild-type cells in Minute cell competition (Kucinski et al. Nature Comm 2017) is a likely example, as JAK/STAT signaling is not known to be mediated by *Xrp1*. As we reported, increased activity of the JAK/STAT pathway, owing to chronic production of *Upd3* in Minute cells, has an effect on competitive growth; to date there is no reason to think that this effect is mediated by *Xrp1*. As there is a sexual dimorphism in circulating *Upd* levels (as reported e.g. in Hudry et al., Cell 2019), this could provide a systemic mechanism for why female loser cells grow better, independently of *Xrp1*. Any other circulating ligand or factor could do the same. Thus, while the finding that *RpS3^{+/-}* sexual dimorphism is partially *Xrp1*-independent is interesting, it does not ultimately change the observation that, in mosaic compartments, *RpS3^{+/-}* cells display sexually dimorphic competitive growth that impacts their tissue colonization abilities in competing settings.

Regardless of the explicit cause of this male/female difference in clone size, we would like to point out that this finding is of substantial technical significance to the field. Prior studies in the field have not generally controlled for gender in their experiments. Given that most labs score clone size as their primary metric of competition (especially as competitive death is manually tedious to score, hence the need for PECAN), even a moderate discrepancy in the proportion of males to females between experimental groups could yield erroneous results, lack of reproducibility and reduced sensitivity.

Furthermore, mutations or experimental interventions where genders are used to select for genotypes (a standard practice in *Drosophila* genetics) would lead to artefactual competition phenotypes. Thus, the finding that there is such a substantial sexual dimorphism in clone size, regardless of cause, is important for improving the quality of future competition experiments and could help explain the notorious lack of reproducibility of some experiments in our field.

Reviewer #2 (Remarks to the Author):

The authors satisfied the critiques and questions raised in the initial submission. The most interesting aspects of the paper are the new quantitative insights gained about cell competition, including sexual dimorphism and the relative importance of stochasticity. If the authors incorporate the essential biological insights into the title it may alert more readers interested in factors impacting cell competition and growth and apoptosis-related phenotypes.

Thank you for recognizing that the work we have carried out satisfies the critiques and questions raised in the initial submission. We agree that revising the title will help alert readers about the competition relevant insights and have edited the title to this effect.

Reviewer #3 (Remarks to the Author):

All my concerns have been addressed. I therefore recommend the paper for publication. However I would like the authors to ensure code is shared (made public) on GitHub.

thank you,
Yanlan Mao

Thank you. The code is now publicly available on Github, and relevant links have been included in the manuscript.

Reviewer #4 (Remarks to the Author):

I really appreciate the effort that the authors put into replying to my previous review comments (e.g. using different flippases, control clones, a different RpS3 competition genetic tool).

As I stated in my previous report the strength of this paper is that it provides a quite useful tool for clonal analysis studies with the potential to be highly appreciated by many researchers and expedite significantly their studies.

Weakness: On the other side, still, in my opinion, this work does not significantly advance our knowledge of the cell competition field, since their conclusion on the role of sexual dimorphism in cell competition is not strongly supported.

We thank the reviewer for appreciating the effort that we have put in addressing their comments and for recognizing that our image and statistical analysis will provide a useful tool to researchers, expediting significantly their studies. The development of this analysis platform is indeed a main resource that we communicate in this paper.

Having said that, in the process of demonstrating the power of PECAn, we have generated important pieces of information that the cell competition community will appreciate, including: a detailed analysis of the stochasticity of competitive cell death; a quantitative measure of competition induced cell death, including the cell range where this is manifest; a RNAi screen with lots of exciting novel cell competition modulators, which we are offering to the community at a stage where they can be taken on for characterization; and the identification of a sexually dimorphic competitive growth phenotype, which we discuss in details further, given this is the major outstanding point for this reviewer.

We believe that the release of PECAn, together with the additional insights on cell competition we describe here, will make this a highly cited and appreciated manuscript.

MAJOR POINTS

My main objection, still, is regarding the proposed role of sexual dimorphism in Minute cell competition. The main reason is the result that gave one of the experiments that I proposed, which in my opinion does not agree with their conclusion. More specifically, Xrp1 is responsible for Minute cell competition, including RpS3 competition. This has been confirmed by many groups, including Dr Piddini's (30078730, 30531963, 31841522, 33495633, 31909714, 34871307, 34914692, 35179490) and it is also shown in Figure 3h, where Xrp1-RNAi completely blocks competitive cell death (Dcp1 positive cells number is zero).

Therefore, since in Xrp1 mutant background the sex difference is still present (line 312-316, Suppl Fig 4c-f), my only interpretation is that this is independent of cell competition. Have the authors performed Dcp1 staining in RpS3-dependent competition in the absence of Xrp1, to check if Dcp1 is still present in males, which could explain the difference in sex?

It has been shown that Xrp1 substantially, but not completely, affects also the growth and the developmental delay of Minutes (PMID: 30078730). Therefore, my interpretation could have been that sex affects the growth of Minutes and not their competition (Meaning the growth of the female Minute cells is higher than males Minute).

The observation that the sexual dimorphism is not suppressed (though it is reduced) by XRP1 heterozygosity is interesting and we thank the reviewer for suggesting that experiment. We have not tested whether Xrp1 affects Dcp1 staining, because the sexually dimorphic phenotype that we observe is not a cell death phenotype, i.e. there is no difference in Dcp1 staining between males and females to start with (shown in Figure 4C). This suggests that this is a competitive growth and not a competitive death phenotype, as indeed the reviewer suggests.

The partial effect of Xrp1 on the sexually dimorphic phenotype is interesting and we believe it does not conceptually rule out the possibility that this is a cell competition phenotype. This can be argued for several reasons.

- 1) As this reviewer points out, Xrp1 strongly, but not completely, suppresses growth differences seen in *Minutes*. Thus, we need not expect that Xrp1 should completely equalize this difference, and we do see a decrease in the difference between male and females in an Xrp1 mutant context (Supplementary Figure 4d, $\Delta = -0.908$ in *RpS3^{+/-}, Xrp1^{+/-}* vs $\Delta = -0.533$ in *RpS3^{+/-}, Xrp1^{+/-}*). Therefore, it is formally possible that some, though probably not all, of the sexually dimorphic phenotype observed is mediated by Xrp1. We have edited the manuscript to point this out.
- 2) We fully agree with the reviewer that this phenotype is a growth phenotype, but growth phenotypes are a component of competition phenotypes. Competitive growth is a parameter of cell competition, like competitive elimination. Like this reviewer, we interpret the data to suggest that growth differences of RpS3 cells, in different sexes, impact the competitive outcome. Still, this is a competition phenotype. We do not suggest that competitive apoptosis is sexually dimorphic. As shown in Figure 3g-h competitive growth and competitive cell death can be modulated independently. Indeed, inhibition of apoptosis reduces competitive cell death without influencing clone size. This finding has also been reported in the literature (Martín FA Herrera SC Morata G, Development 2009) In the case of the sexual dimorphism observed, it is likely that the reverse is happening, and that growth rather than death is affected. We have revised the text to clarify this point.
- 3) Genes and pathways known to influence *Minute* cell death and growth in a cell autonomous context are accepted as modulators of Minute cell competition. This is true of pathways ameliorating proteostasis, for example (Baumgartner et al, NCB 2021; Recasens-Alvarez, NCB 2021). The role of Xrp1 itself in competition runs into a similar issue of interpretation. Xrp1 knockdown rescues proteotoxic stress, cell growth, cell death, and stress pathway activation autonomously in *Minutes*. This leads to the question: does Xrp1 suppress competition or is the rescue of clone size simply a reflection of a cell autonomous rescue of growth? does it suppress competitive death or does it rescue the cell autonomous stressors that predispose cells to be outcompeted? For both questions, the latter interpretation is the currently accepted, which points at Xrp1 having an indirect effect on the competition process. Despite this, no one would deny that Xrp1 knockdown yields a competition phenotype. By the same token, sexually dimorphic mechanisms affecting autonomous cell growth should be accepted as mechanisms that impact on competitive growth.
- 4) Importantly, while it is established that Xrp1 is necessary for Minute competition, this does not rule out that additional, Xrp1-independent mechanisms, may also be at play. For example, as we reported in Kucinski et al., 2017, the JAK/STAT pathway, activated by chronic production of Upd3 in Minute cells, has an effect on competitive growth; to date there is no reason to think that this effect is mediated by Xrp1. Similarly, sexually dimorphic availability of Upd (as reported e.g. in Hudry et al., Cell 2019) or other ligands may preferentially affect wild-type or RpS3 cells during cell competition, in an Xrp1-independent manner.

Finally, we would like to point out that, regardless of the explicit cause of this male/female difference in clone size, this finding is of substantial technical significance to the field. Prior studies in the field have not generally controlled for gender in their experiments. Given that most labs score clone size as their primary metric of competition (especially as competitive death is manually tedious to score, hence the need for PECA_n), even a moderate discrepancy in the proportion of males to females between experimental groups could yield erroneous results, lack of reproducibility and reduced sensitivity. Furthermore, mutations or experimental interventions where genders are used to select for genotypes (a standard practice in *Drosophila* genetics) would lead to artefactual competition phenotypes. Thus, the

finding that there is such a substantial sexual dimorphism in clone size, regardless of cause, is important for improving the quality of future competition experiments and could help explain the notorious lack of reproducibility of some experiments in our field.

In sum, we find ourselves overall in agreement with the reviewer's view that this is most likely a competitive growth phenotype and that it does not necessarily reflect an explicit winner/loser interaction and we have amended the text to that effect.

On the other hand, it has been shown that by the end of larval development wing discs of wild-type females are larger than that of wild-type males (for example in Figure 6 of PMID: 28976974), supporting the increased larval growth in females that has been shown previously. So, I do not feel comfortable even stating a clear conclusion that the increased cellular growth in females is due to Minute mutation.

This is a reasonable criticism which we have considered ourselves, however several of our observations help us to rule this possibility out.

- In the scenario suggested by the reviewer, *Minute* clones would get bigger in absolute terms in females, but so would wild-type clones, so pouch coverage, our measure of competition, which is a % value, would not be affected, unlike we observe. The fact that it is indicates that this sexually dimorphic growth is specific to Minute cells and/or to Minute cell competition. Indeed, in the experiments where we generated clones using the MiWO construct, but in a wildtype rather than in a $RpS3^{+/-}$ background, we observed no sexual dimorphism in clone size. This argues against the possibility of a general effect on clone size due to female wing discs growing bigger. It also rules out that a 24-hour collection window can account for this phenotype, as this timing was not an issue for experiments conducted using the same construct.
- The experiment in Figure 4f-g also argues against the hypothesis of a simple difference in wing disk size between males/females as the explanation. Here, rather than using the MiWO construct, we used classic FRT-mediated mitotic recombination. In this set up the starting genotype is $RpS3^{+/-}$ and wild-type clones are introduced in it; in the MiWO set up, the opposite is true. Thus, a simple scenario wherein the clones introduced are allowed to grow bigger in females than in males does not hold, because in the former genetic set up it would be the wild-type clones to grow bigger in females, whereas in the MIWO set up, it would be the $RpS3^{+/-}$ clones to grow bigger. The fact that in both experiments we end up with more $RpS3^{+/-}$ cells in females indicates that female/male differences are specifically affecting $RpS3^{+/-}$ cells and not all cells.

My concern is that maybe other parameters affect the sex difference in clonal size that is observed in Minute clones that are induced in wild-type tissue, but not in wild-type clones that are in wild-type tissue. Given the 24hr period of egg laying and the differential larval growth of males and females, for me, it is hard to draw straightforward conclusions.

We fully agree with this statement, but we consider this a valid mechanism that would meet the criteria of sexually dimorphic effect on cell competition. Indeed, as we point out above, *Minute* cell competition involves both growth phenotypes and cell death phenotypes. In addition, a competition phenotype can be influenced by systemic factors rather than by explicit winner/loser interactions (e.g. circulating levels

of insulin influence *Scribble* competition, as reported by Sanaki et al., Dev. Cell. 2020). Thus, something which 'affect[s] the sex difference in clonal size that is observed in Minute clones that are induced in wild-type tissue, but not in wild-type clones that are in wild-type tissue' would meet the definition of competition.

In addition, having only one different Minute mutation, someone could argue that this has something to do with a specialized role of RpS3 protein and is not a general effect of Rp mutant cells. I indeed agree that they revised a significant part of the paper, but including a different Minute (for example RpL14) could have strengthened their conclusion, even regarding the role of sex in the growth of Minute cells.

We thank the reviewer for raising this valid point, which we also felt was important to address. For this reason, prior to receiving this comment and while the manuscript was under review, we set up experiments to address this. We are pleased to report that a dimorphic effect is also observed with a different Minute, specifically *RpL27A*. The new data have been added to the manuscript and can be found in Figure 4h-i.

Unfortunately, I can not agree that there is a substantial contribution in the cell competition field and my suggestion is that this work might be better suited in a more specialized journal.

Best regards.

We hope that the presented arguments above and the new data and edits we have made to the manuscript will overcome the remaining concerns of this reviewer.

Reviewers' Comments:

Reviewer #1:

Remarks to the Author:

The manuscript has been greatly improved, and I heartily thank the authors for putting in so much effort to make it more clear and answer reviewer queries.

However, I do still have issues with their conclusions about the sexual dimorphism they associate with Minute cell competition. I tend to agree with Reviewer 4 that given the 24 hour egg lay duration (which I also remarked on previously), it is difficult to solidly conclude much about dimorphic disc growth, either in cell competition or in the Minute background itself. I agree with the authors that examining males and females separately is an important goal for these types of competitive studies and think their raising this topic is important, so perhaps being more speculative about their conclusions in this regard would suffice. They might want to see Svoisky et al 2021, PMID 34909609, which also examined sexual dimorphism in cell competition.

Reviewer #4:

Remarks to the Author:

I would like to thank the authors for the new data provided in this revised version. The reasons that I am very critical in the sexual dimorphism in growth of competing Minute cells are the following:

a) for the cell competition field, I believe this the most interesting part of this work.
b) since males and females do not affect competitive death, that means that for the previous studies that used the competitive cell death as proxy for competitiveness, their conclusions are still valid even if they used mixed populations. Only the studies that used solely the clonal size as a proxy for cell competition should be re-evaluated, in case that they used mixed populations. I am not sure that this will explain the notorious lack of reproducibility of some experiments in our field, since in most of the cases the cell death is the most important characteristic of Minute competition that people examine.

c) Since this will affect also the future studies, we have to be careful to exclude other reasons for these differences (e.g. technical, developmental delay, time of inducing clones).

Regarding the revised version of this work my comments are the following.

It is indeed positive that RpL27 Minutes present the same dimorphic effect, supporting the non-RpS3 specific response. The authors used a different RpL27 Minute mutation (RpL27A1), which even if it gave similar results regarding p-eIF2a with the one (Df(2 L)M24F11) that a previous study used (PMID: 35179490), it showed different results regarding the p62 accumulation. This is interesting, since both RpL27A mutations make cells losers (even if the Df deletes more genes). Therefore, I do not agree that this is a strong argument that proteotoxic stress is a general feature of loser phenotypes.

The authors state on their rebuttal: "Our reported role of JAK/STAT in the competitive growth of wild-type cells in Minute cell competition (Kucinski et al. Nature Comm 2017) is a likely example, as JAK/STAT signaling is not known to be mediated by Xrp1. As we reported, increased activity of the JAK/STAT pathway, owing to chronic production of Upd3 in Minute cells, has an effect on competitive growth; to date there is no reason to think that this effect is mediated by Xrp1. As there is a sexual dimorphism in circulating Upd levels (as reported e.g. in Hudry et al., Cell 2019), this could provide a systemic mechanism for why female loser cells grow better, independently of Xrp1." Actually it was shown by Baker's lab that Upd3 expression in both RpS17 and RpS3 Minutes depends on Xrp1 (Fig. 1C, 1H, 1I in PMID: 31841522). Also, JNK activation, which was shown by Piddini's lab to contribute to JAK/STAT ligands production in Minute cells, has been shown to depend on Xrp1 expression (PMID: 30078730, 31841522, 35179490).

In addition, Xrp1 indeed autonomously reduces the growth of Minutes and is responsible for many other stress responses in Minutes (e.g. reduced translation, p-eIF2a accumulation, JNK activation, p62 accumulation, Upd3 expression), but clearly Xrp1 depletion completely rescues the Minutes from competitive cell death when are in mosaics with wild type cells (Dcp1 of Minutes cells at boundaries with wild type cells goes down to zero, if Minute cells lack Xrp1). It is different from the sexual dimorphic effect, where competitive cell death (the main characteristic of Minute competition as they also state in line 86-88) is not affected by sex.

If the sex differences in growth are intrinsic to Minutes and do not depend on the presence of wild type cells, they should not mention it as competitive growth phenotype, just sexual dimorphism in Minute phenotype.

Lastly, the authors by statistical regression analysis identified tissue parameters that model and correlate with competitive death. One of these parameters is the anti-correlation of the predicted probability of apoptosis with the total number of losers cells (Figure 6b), which is in agreement with other competition studies. Having this parameter in mind, if we try to interpret the result shown in Figure 4a, 4b and 4c we will come to the conclusion that the competition in males is reduced, since even if total number of losers in males in 4a,4b is less than females, the males do not present higher competitive apoptosis as predicted from their model (the apoptosis is the same in both males and females in 4c). This conclusion is opposite to the previous one that in males Minute competition has disadvantage. How this is explained?

I find essential for the competition field to clarify the above points in the paper. Especially for the fact that the authors have not proven that the sexual dimorphism of growth depend on competition with wild type cells and not just an autonomous effect on their growth. I still have the concern, regarding the 24 hours egg collection, but since the rest of the experiments show some effect of sex in growth of Minutes, my concerns are less.

Thank you

REVIEWER COMMENTS

Reviewer #1 (Remarks to the Author):

The manuscript has been greatly improved, and I heartily thank the authors for putting in so much effort to make it more clear and answer reviewer queries.

However, I do still have issues with their conclusions about the sexual dimorphism they associate with Minute cell competition. I tend to agree with Reviewer 4 that given the 24 hour egg lay duration (which I also remarked on previously), it is difficult to solidly conclude much about dimorphic disc growth, either in cell competition or in the Minute background itself. I agree with the authors that examining males and females separately is an important goal for these types of competitive studies and think their raising this topic is important, so perhaps being more speculative about their conclusions in this regard would suffice. They might want to see Svoisky et al 2021, PMID 34909609, which also examined sexual dimorphism in cell competition.

Thank you for these constructive comments. We have added citations to Svoisky et al 2021 and rephrased our statements to be more speculative.

We nonetheless wish to highlight, regarding the 24hour egg collection, that one can always argue whether the experimental conditions chosen (e.g. for egg collection and for time of inducing clones and al dissection) are absolutely perfect. However, the fact that using these same conditions, all of our negative controls failed to give a sexual dimorphism, whereas it showed in the Minute experimental data sets for multiple genotypes, allows us to univocally conclude that this phenotype is specific to the clonal growth of ribosome mutant cells. In fact, I would argue the opposite, i.e. that given that we have observed it with a relaxed egg collection window, we would most certainly, and even more strongly detect it with a narrower one, as this would have only increased our sensitivity.

Reviewer #4 (Remarks to the Author):

I would like to thank the authors for the new data provided in this revised version. The reasons that I am very critical in the sexual dimorphism in growth of competing Minute cells are the following:

- a) for the cell competition field, I believe this the most interesting part of this work.
- b) since males and females do not affect competitive death, that means that for the previous studies that used the competitive cell death as proxy for competitiveness, their conclusions are still valid even if they used mixed populations. Only the studies that used solely the clonal size as a proxy for cell competition should be re-evaluated, in case that they used mixed populations. I am not sure that this will explain the notorious lack of

reproducibility of some experiments in our field, since in most of the cases the cell death is the most important characteristic of Minute competition that people examine.

We agree wholeheartedly with this conclusion. We hope with this manuscript to improve reproducibility via two means.

Firstly, our observation of clone size differences between genders raises a red flag for previous work, where clone size only was used as competition parameter and sets new standards for future work, emphasizing the importance of scoring for competitive cell death. This is especially important, as experiments assessing clone size remain one of the most common metrics for evaluating competition phenotypes in the field.

Second, for experiments quantifying cell death, the PECAN tool will improve consistency and reduce inter-operator variability in quantification of experiments.

c) Since this will affect also the future studies, we have to be careful to exclude other reasons for these differences (e.g. technical, developmental delay, time of inducing clones).

We are confident that our thorough experiments have ruled out all possible confounding factors. Importantly, while one can always argue whether the experimental conditions chosen (e.g. for egg collection and for time of inducing clones and al dissection) are absolutely perfect, the fact that using these same conditions in all of our negative controls failed to give a sexual dimorphism, whereas it did in the experimental data sets for multiple genotypes (across two different operators), allows us to univocally conclude that this phenotype is specific to the clonal growth of ribosome mutant cells.

Regarding the revised version of this work my comments are the following.

It is indeed positive that RpL27 Minutes present the same dimorphic effect, supporting the non-RpS3 specific response. The authors used a different RpL27 Minute mutation (RpL27A1), which even if it gave similar results regarding p-eIF2a with the one (Df(2L)M24F11) that a previous study used (PMID: 35179490), it showed different results regarding the p62 accumulation. This is interesting, since both RpL27A mutations make cells losers (even if the Df deletes more genes). Therefore, I do not agree that this is a strong argument that proteotoxic stress is a general feature of loser phenotypes.

This is a valid criticism, and we have toned down our language to say "our data showing that it is also found in *RpL27A* cells argue that proteotoxic stress is not exclusively a phenotype of small ribosomal subunit mutations.."

The authors state on their rebuttal: "Our reported role of JAK/STAT in the competitive

growth of wild-type cells in Minute cell competition (Kucinski et al. Nature Comm 2017) is a likely example, as JAK/STAT signaling is not known to be mediated by Xrp1. As we reported, increased activity of the JAK/STAT pathway, owing to chronic production of Upd3 in Minute cells, has an effect on competitive growth; to date there is no reason to think that this effect is mediated by Xrp1. As there is a sexual dimorphism in circulating Upd levels (as reported e.g. in Hudry et al., Cell 2019), this could provide a systemic mechanism for why female loser cells grow better, independently of Xrp1." Actually it was shown by Baker's lab that Upd3 expression in both RpS17 and RpS3 Minutes depends on Xrp1 (Fig. 1C, 1H, 1I in PMID: 31841522). Also, JNK activation, which was shown by Piddini's lab to contribute to JAK/STAT ligands production in Minute cells, has been shown to depend on Xrp1 expression (PMID: 30078730 , 31841522, 35179490).

These points are all very relevant and correct but are not incompatible with our argument. Even though, as the reviewer states, XRP1 is *upstream* of both JNK and Upd3 production, we argue that the signaling response *downstream* of Upd3 is likely XRP1-independent and could be sexually dimorphic (as could the production of Upd3 ligands), possibly contributing to sexually dimorphic growth.

In addition, Xrp1 indeed autonomously reduces the growth of Minutes and is responsible for many other stress responses in Minutes (e.g. reduced translation, p-eIF2a accumulation, JNK activation, p62 accumulation, Upd3 expression), but clearly Xrp1 depletion completely rescues the Minutes from competitive cell death when are in mosaics with wild type cells (Dcp1 of Minutes cells at boundaries with wild type cells goes down to zero, if Minute cells lack Xrp1). It is different from the sexual dimorphic effect, where competitive cell death (the main characteristic of Minute competition as they also state in line 86-88) is not affected by sex.

This is indeed correct: competitive cell death is not affected by sex, but competitive growth is.

If the sex differences in growth are intrinsic to Minutes and do not depend on the presence of wild type cells, they should not mention it as competitive growth phenotype, just sexual dimorphism in Minute phenotype.

We agree with this point and currently cannot distinguish between these two possibilities. Accordingly, we have toned down our language and added caveats in the manuscript where appropriate. However, we would like to emphasize that regardless of whether this is a consequence of competition with wild-type cells or a cell-autonomous feature of Minute cells, it impacts substantially competitive growth. Thus, for all the reasons indicated by the

reviewer in points a), b), c) above, it has a big impact both in the manifestation of cell competition and in how investigators should score cell competition.

Lastly, the authors by statistical regression analysis identified tissue parameters that model and correlate with competitive death. One of these parameters is the anti-correlation of the predicted probability of apoptosis with the total number of losers cells (Figure 6b), which is in agreement with other competition studies. Having this parameter in mind, if we try to interpret the result shown in Figure 4a, 4b and 4c we will come to the conclusion that the competition in males is reduced, since even if total number of losers in males in 4a,4b is less than females, the males do not present higher competitive apoptosis as predicted from their model (the apoptosis is the same in both males and females in 4c). This conclusion is opposite to the previous one that in males Minute competition has disadvantage. How this is explained?

This is indeed very logical, why if clones tend to be smaller in males, we do not see a higher amount of cell death? Does this mean that in fact competitive cell death is comparatively less strong in males than in females? This is an example of the types of questions that PECAN will enable the field to address once it is released. Indeed, while it would not be possible to extract this type of answers from standard ways of analyzing competitive death this will be possible with PECAN. A key difference in the experiments run in Figure 4a, 4b and 4c and in (Figure 6b), is in the vast difference in sample size. The regression analysis in Figure 6 is the result of a high-powered investigation of individual cells with a sample size of close to 200,000 cells. The second regression analysis involves counts of hundreds of thousands of apoptotic vs non-apoptotic cells across 183 wing discs. Thus, these experiments are of a vastly higher statistical power, and findings which are evident there need not be evident in a lower powered experiment comparing densities of dying cells in comparably small numbers of wing discs, such as in Figure 4. Thus, it could either be the case, as the reviewer suggests, that in fact competitive death is weaker in males, or simply that the lower powered experiments in Figure 4 cannot pick up a subtler effect, predicted on the basis of our higher-powered analysis in Figure 6. As we did not dissect and analyze a large number of male wing discs in this regression analysis, we cannot conclude anything further based on the data available.

I find essential for the competition field to clarify the above points in the paper. Especially for the fact that the authors have not proven that the sexual dimorphism of growth depend on competition with wild type cells and not just an autonomous effect on their growth. I still have the concern, regarding the 24 hours egg collection, but since the rest of the experiments show some effect of sex in growth of Minutes, my concerns are less.

Thank you

We thank the reviewer for their thorough analysis and have included the suggested clarifications in our manuscript.